# ALKBH5-mediated m6A modification of IL-11 drives macrophage-to-myofibroblast transition and pathological cardiac fibrosis in mice

Tao Zhuang[1,10], Mei-Hua Chen[1,2,10], Ruo-Xi Wu[1], Jing Wang[1], Xi-De Hu[1], Ting Meng[1], Ai-Hua Wu[3], Yan Li[4], Yong-Feng Yang[1], Yu Lei[1], Dong-Hua Hu[1], Yan-Xiu Li[5], Li Zhang[6], Ai-Jun Sun [7], Wei Lu [3] ✉, Guan-Nan Zhang[8] ✉, Jun-Li Zuo[9] ✉ & Cheng-Chao Ruan [1] ✉

Cardiac macrophage contributes to the development of cardiac fibrosis, but factors that regulate cardiac macrophages transition and activation during this process remains elusive. Here we show, by single-cell transcriptomics, lineage tracing and parabiosis, that cardiac macrophages from circulating monocytes preferentially commit to macrophage-to-myofibroblast transition (MMT) under angiotensin II (Ang II)-induced hypertension, with accompanying increased expression of the RNA N6-methyladenosine demethylases, ALKBH5. Meanwhile, macrophage-specific knockout of ALKBH5 inhibits Ang II-induced MMT, and subsequently ameliorates cardiac fibrosis and dysfunction. Mechanistically, RNA immunoprecipitation sequencing identifies interlukin-11 (IL-11) mRNA as a target for ALKBH5-mediated m6A demethylation, leading to increased IL-11 mRNA stability and protein levels. By contrast, overexpression of IL11 in circulating macrophages reverses the phenotype in ALKBH5-deficient mice and macrophage. Lastly, targeted delivery of ALKBH5 or IL-11 receptor α (IL11RA1) siRNA to monocytes/macrophages attenuates MMT and cardiac fibrosis under hypertensive stress. Our results thus suggest that the ALKBH5/IL-11/IL11RA1/MMT axis alters cardiac macrophage and contributes to hypertensive cardiac fibrosis and dysfunction in mice, and thereby identify potential targets for cardiac fibrosis therapy in patients.

Heart failure (HF), characterized by pathological cardiac fibrosis and hypertrophy, is an important cause of morbidity and mortality worldwide[1]. In fact, cardiomyocytes only account for approximately 30% of the cells in the heart; non-myocytes, including endothelial cells, fibroblasts and immune cells, are the most abundant cell types[2,3]. Among these, cardiac fibrosis is characterized by fibroblast activation and differentiation into myofibroblasts, that produce excessive extracellular matrix deposition[4]. Accumulating evidence has emerged on the role of a pro-inflammatory state as a major determinant of cardiac fibrosis pathophysiology. Coordinated innate and adaptive immune responses culminate in sequential immune cell infiltration into the heart contributing to cardiac inflammation and fibrosis[5]. Recent advances in single-cell transcriptomic technologies (scRNA seq) have revealed the variety of immune cells in the heart adding many levels of complexity to our understanding of cardiac cell identity[6,7]. Innate immune cells, especially macrophages have been

well-documented to be implicated in the development of cardiac fibrosis[8].

Cardiac macrophage mainly arises from embryonic development and fetal monocytes and circulating monocytes[9]. Cardiac macrophages derived from embryonic primitive yolk-sac and fetal monocyte progenitor express chemokine receptor Cx3cr1, with low surface expression of CCR2[9], and persist into adulthood but the initially high contribution to resident cardiac macrophage decline after birth, and are progressively substituted by monocyte-derived CCR2 highly expressed macrophages in the adulthood. Genetic fate mapping analysis revealed that all adult cardiac macrophages develop from Cx3cr1[+] progenitors. Previous studies have shown robust tamoxifen-inducible reporter gene expression in Cx3cr1 expressing cardiac resident macrophages and transient labeling of blood monocytes[10,11]. Cx3cr1 cardiac macrophages are required for preservation of cardiac function and fibrosis during hypertensive stress and myocardial infarction[11-13]. Although the sources and progenitors of cardiac resident and infiltrated macrophages have been well documented, the factors that regulate the phenotype or transition and activation of these macrophages in the heart remain to be illustrated during cardiac fibrosis progression.

In this work, we show that cardiac macrophages have a tendency to differentiate into myofibroblasts with accompanying increased RNA N6-methyladenosine demethylase alkylation repair homolog protein 5 (ALKBH5) expression. Specific deletion of ALKBH5 in macrophage inhibits hypertension or pressure overload induced MMT, and subsequently attenuates pathological cardiac fibrosis and diastolic dysfunction, via regulating m6A modification on IL-11 mRNA. Nanoparticle macrophage-target delivery of ALKBH5 or IL11RA1 siRNA attenuates Ang II-induced cardiac fibrosis and diastolic dysfunction. In summary, our study demonstrates that mouse cardiac macrophages from circulating monocytes may trans-differentiate into myofibroblast under hypertensive conditions for fibrosis development, with an AKLBH5/IL-11 molecular axis modulating this macrophage-to-myofibroblast transition.

## Results

### Hypertension increases macrophage-myofibroblast transition

We firstly sorted cardiac non-myocardial cells from Angiotensin II (Ang II) infused C57BL/6 mice to perform single-cell RNA sequencing (scRNA sequencing) assay (Fig. 1A). Cell clusters were identified using UMAP dimensionality reduction analysis based on marker gene expression. We found 9 clusters, including fibroblast, myofibroblast, macrophage, endothelial cells (EC), T&NK, B cell, smooth muscle cell (SMC), myocardial cell (CM), and a small unknown cluster (X) (Fig. 1B, Supplementary Fig. 1A). To investigate the roles of macrophages in hypertension-induced cardiac dysfunction, we further clustered macrophages, which expressed CD68, Adgre1, Fcgr1 and Csf1r, into 15 small populations (Fig. 1C, Supplementary Fig. 1B-1C). These populations could be defined as 2 macrophage types: circulating monocyte-derived macrophages (Ccr2+ and Il1β+), and cardiac resident macrophages (Lyve1+ and Timd4+). Interestingly, scRNA-seq showed a small population of circulating monocyte-derived macrophage expressed Acta2 (SMA) and Tagln, that were typical myofibroblast markers (Fig. 1D). We then used scVelo[14] to visualize RNA velocity, and the pseudotime tranjectory inferred that there may be a link between circulating monocytes-derived macrophages and myofibrobalsts (Fig. 1E, Supplementary Fig. 1D).

To confirm the existence of MMT in monocytes/macrophages during hypertensive stress, we performed lineage tracing assay, including Cx3cr1Cre; Rosa26[Td] mice[10], Lyz2Cre[15]; Rosa26[Td] mice, and CCR2-GFP reporter mice[16] to label circulating monocyte-derived macrophages; Cx3cr1Cre[ERT2];Rosa26[Td] mice[12] and Lyve1Cre;Rosa26[Td] mice[11] to label cardiac resident macrophages. FACS analysis in hearts from

Cx3cr1Cre; Rosa26[Td] mice showed few SMA+CD11b+Td+ cells after PBS treatment, while a cluster of SMA+CD11b+Td+ cells following Ang II treatment (Fig. 1F). Immunostaining also revealed an increased SMA +Td+ myofibroblasts generation in the hearts of Ang II-infused Cx3cr1Cre; Rosa26[Td] mice (Fig. 1G). In consistent with Cx3cr1Cre; Rosa26[Td] mice, we also observed increased SMA+Td+ cells in the hearts of Ang II-infused Lyz2Cre; Rosa26[Td] mice, as well as SMA + GFP+ cells in the hearts of CCR2GFP mice after Ang II-infusion (Fig. 1F, G). Moreover, Ang II treatment induced SMA+Td+ myofibroblast-like cell formation in primary cultured macrophages from Cx3cr1Cre; Rosa26[Td] or Lyz2Cre; Rosa26[Td] mice (Supplementary Fig. 2A–D). Consistently, in vitro cell tracking in CCR2GFP mice showed that Ang II treatment induced GFP+ macrophage transition to SMA + GFP+ myofibroblast-like cells (Supplementary Fig. 2B). In contrast, we did not observe cardiac resident macrophage transition to myofibroblast after Ang II-infusion by utilizing Cx3cr1Cre[ERT2];Rosa26[Td] and Lyve1Cre;Rosa26[Td] mice (Supplementary Fig. 3).

Smooth muscle cells (SMCs) are reported to upregulate macrophage-associated gene programs during inflammatory responses[17]. To exclude the potential hypothesis that smooth muscle cells upregulated macrophage-associated gene programs under Ang II treatement, Myh11Cre[ERT2]; Rosa26[Td] mice were utilized to test whether SM cells upregulated macrophage-associated gene programs. FACS and immunostaining revealed Td+SMA+ cells, but not Td+CD11b cells were increased in hypertensive hearts (Supplementary Fig. 3E–G). Although we observed an increased CD11b+ cells in hypertensive hearts, these cells were not derived from Td+ cells. We then performed parabiosis experiments by conjoining non-fluorescent C57BL6/J mice with Myh11Cre[ERT2]; Rosa26[Td] mice and observed few Td+ cells in hearts from both PBS and Ang II treated C57BL6/J mice assessed by FACS analysis (Supplementary Fig. 3H), indicating that Myh11+ SM cells in the heart do not contribute to macrophage-like cells in this pathological process. Taken together, these indicate that hypertension induces circulating monocyte-derived macrophages, but not cardiac resident macrophages, to transition to myofibroblasts.

### ALKBH5 upregulation is involved in hypertension-induced cardiac MMT

To determine the molecular mechanism that is involved in the regulation of MMT and pathological cardiac fibrosis in the heart, we re-analyzed the scRNA sequencing profiles. Bioinformatic analysis showed that RNA N6-methyladenosine demethylase ALKBH5 was widely expressed in all clusters in the heart with more variation in macrophage cluster (Supplementary Fig. 4A). We next analyzed the ALKBH5 expression in the macrophage subclusters, and observed a higher ALKBH5 expression in myofibroblasts compared with macrophage populations shown by dot plots (Fig. 2A), heat maps (Fig. 2B). Pseudotime analysis further suggested that ALKBH5 expression level was in parallel with the expression of Acta2 (Fig. 2C). The increased ALKBH5 expression during the process of MMT was also shown by Vilon image (Fig. 2D).

N6-methyladenosine (m6A) RNA modification, which are mediated by readers (Ythdc1, Ythdc2, Ythdf1, Ythdf2, Ythdf3 and Hnrnpa2b), writers (Mettl3, Mettl14 and Wtap) and eraser proteins (Fto and ALKBH5), plays various roles in mRNA instability, translation, splicing and phase separation[18]. m6A modification was reported to control cardiac homeostasis and hypertrophy[19]. To define whether m6A modification participates in macrophage-myofibroblast transition under hypertensive stress, we performed m6A dot blot analysis and found that m6A methylation was significantly decreased in macrophages after Ang II treatment (Fig. 2E). We also analyzed the other m6A modification proteins in the scRNA sequencing data, and found that except for ALKBH5, the others had no significant correlation to the

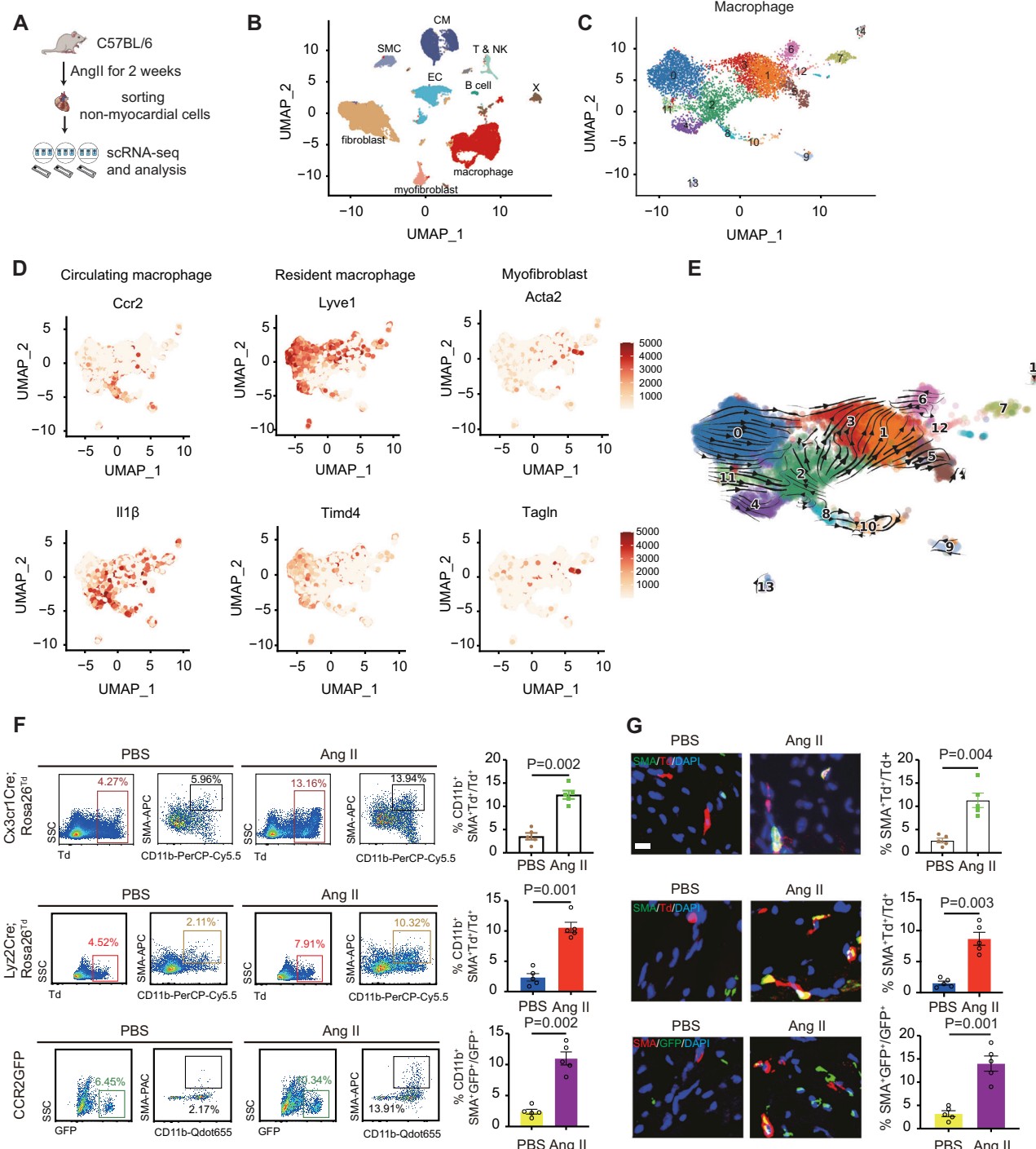

**Fig. 1 | Hypertension increases macrophage-myofibroblast transition.**
**A** Diagram of single cell RNA sequencing of non-myocardial cells from Angiotensin II or PBS infused C57BL/6 mice. **B** UMAP dimensionality reduction analysis displayed clusters of non-myocardial cells. **C** UMAP dimensionality reduction analysis of macrophage. **D** Feature plots depicting single-cell gene expression of selected macrophage and myofibroblast genes by UMAP dimensionality reduction analysis. **E**, RNA velocity (scVelo) analysis showed macrophage to myofibroblast transition. **F** Representative flow cytometry analyses of cardiac CD11b⁺SMA⁺ cells gated on Td⁺ cells from PBS or Ang II treated *Cx3cr1Cre; Rosa26^Td* and *Lyz2Cre;Rosa26^Td* mice, or

GFP+ cells from PBS or Ang II treated *CCR2GFP* mice, with representative images at left and quantification at right. *n* = 5. **G** Representative immunofluorescent images (left) and quantification (right) of SMA⁺Td⁺ cells of cardiac tissues from *Cx3cr1Cre; Rosa26^Td*, *Lyz2Cre; Rosa26^Td* and *CCR2GFP* mice with and without Ang II infusion. *n* = 5. Scale bar, 50 μm. All data are presented as mean ± standard error mean. Data in **F**, **G** were analyzed by two-tailed unpaired Student's *t*-test. *P* < 0.05 was considered as statistically significant. Ang II Angiotensin II, scRNA-seq single cell RNA sequencing, UMAP uniform manifold approximation and projection, SMC smooth muscle cell, CM cardiomyocyte, EC endothelial cell, NK natural killer.

MMT process (Supplementary Fig. 4B). We also assessed the mRNA level of m6A modification proteins in sorted Td+ cells from *Cx3cr1Cre;Rosa26^Td* mice by RT-qPCR, and found that ALKBH5 notably increased after AngII-infusion (Supplementary Fig. 4C). We also

observed an upregulated ALKBH5 protein level in vitro cultured Td+ macrophages following Ang II treatment (Fig. 2F). We therefore hypothesized that ALKBH5 might contribute to hypertension-induced MMT, and related cardiac fibrosis and dysfunction.

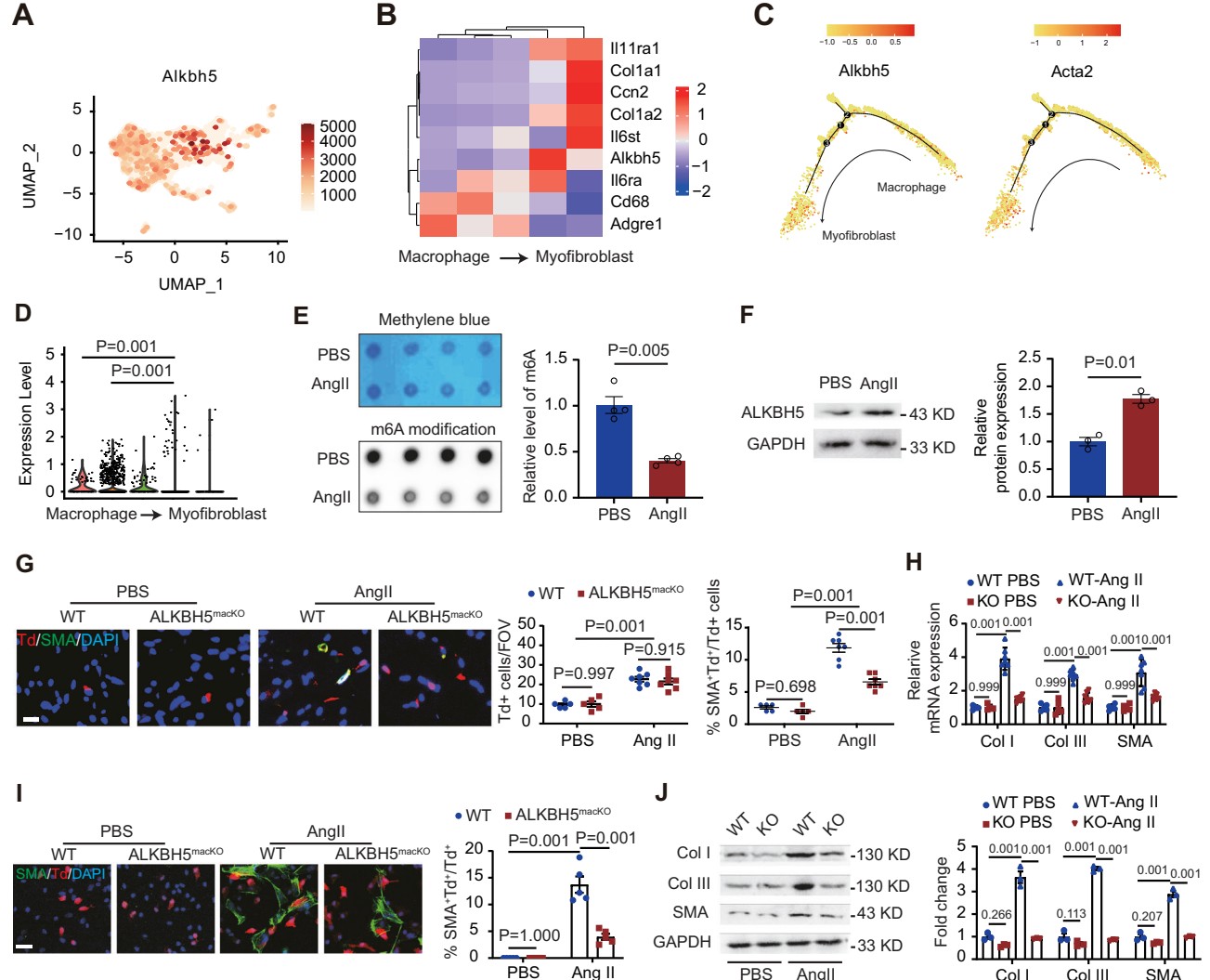

**Fig. 2 | ALKBH5 upregulation is involved in hypertension-induced cardiac MMT. A** Dot plots depicting single-cell gene expression of ALKBH5 in macrophage subclusters visualized on the UMAP dimensionality reduction plots. **B** The heat maps showing the marker gene and ALKBH5 expression of each cell cluster. **C** The ALKBH5 and Acta2 expression in pseudotemporal trajectory analysis from cluster macrophages to myofibroblasts. **D** ALKBH5 expression in macrophages and myofibroblast-like macrophages shown by violin. **E** m6A dot blot assay of mRNAs in cardiac Td+ cells from *Cx3cr1Cre$^{Td}$* mice with or without Ang II infusion (*n* = 5). **F** Expression of ALKBH5 by western blot in cardiac Td$^+$ cells from *Cx3cr1Cre$^{Td}$* mice with and without Ang II infusion. *n* = 5. **G** Representative immunofluorescent images and quantification of SMA$^+$Td$^+$ cells in cardiac tissues from *Cx3cr1Cre$^{Td}$* and *ALKBH5$^{macKO-Td}$* mice (*n* = 5 for PBS group and *n* = 7 for Ang II group). Scale bar, 100 μm. **H** mRNA levels of SMA and collagen (Col) types I and III expression in cardiac Td$^+$ cells from *Cx3cr1Cre$^{Td}$* and *ALKBH5$^{macKO}$* mice. *n* = 5 for PBS group and *n* = 7 for

Ang II group. **I** Representative immunofluorescent images (left) and quantification (right) of SMA$^+$ cells in cultured Td$^+$ macrophages cells from *Cx3cr1Cre$^{Td}$* and *ALKBH5$^{macKO}$* mice with and without Ang II infusion. *n* = 5. Scale bar, 100 μm. **J** Representative images and quantification of the SMA and collagen (Col) types I and III expression by western blot in cardiac tissues of *Cx3cr1Cre$^{Td}$* and *ALKBH5$^{macKO-Td}$* mice (*n* = 5 for PBS group and *n* = 7 for Ang II group). All data are presented as mean ± standard error mean. Data in **E**, **F** were analyzed by two-tailed unpaired Student's t test. Data in **D** were analyzed by Wilcoxon signed-ranks test. Data in **G** were analyzed by two-way ANOVA followed by Tukey post-hoc tests. Data in **H**–**J** were analyzed by one-way ANOVA followed by Tukey post-hoc tests. n.s. indicates nonsignificant. *P* < 0.05 was considered as statistically significant. UMAP uniform manifold approximation and projection, Ang II Angiotensin II, FOV field of view, WT wild-type, KO knockout, Col collagen.

## Specific deletion of ALKBH5 in Cx3cr1 lineage improves Ang II- or pressure overload-induced MMT and cardiac fibrosis

We next constructed *Cx3cr1cre; Rosa26$^{Td}$; ALKBH5$^{flox/flox}$* (*ALKBH5$^{macKO-Td}$*, KO) mice to investigate the role of macrophage ALKBH5 in MMT and pathological cardiac fibrosis. The expression of ALKBH5 was significantly decreased in peritoneal macrophages of *ALKBH5$^{macKO-Td}$* mice compared with control mice (*Cx3cr1cre; Rosa26$^{Td}$; ALKBH5$^{wt/wt}$*, WT) (Supplementary Fig. 5A, RIP-sequencing Dataset). We firstly determined the effect of ALKBH5 deletion on MMT in vivo and in vitro. ALKBH5 deletion did not change Td+ cells in hearts under steady state or following challenge (Fig. 2G). *ALKBH5$^{macKO-Td}$* mice displayed a significant decrease in SMA+Td+ cells in the hypertensive hearts (Fig. 2G),

as well as reduction of SMA and ECM gene collagen I and III mRNA level in sorted Td+ cells (Fig. 2I). Consistent with in vivo findings, we found that Ang II increased but ALKBH5 knockout reversed the increased SMA+Td+ myofibroblast-like cell formation and expression of SMA and ECM genes collagen I and III in cultured macrophages after Ang II treatment (Fig. 2I, J).

We then determined the effect of macrophage ALKBH5 deletion on cardiac function and fibrosis. We found that ALKBH5 deletion had no significant effect on Ang II-induced blood pressure elevation (Fig. 3A). Ang II-infusion for 14 days displayed no significant ejection fraction (EF) decrease, but E/e' ratio increase, indicating that hypertension induced cardiac diastolic dysfunction. *ALKBH5$^{macKO-Td}$* mice

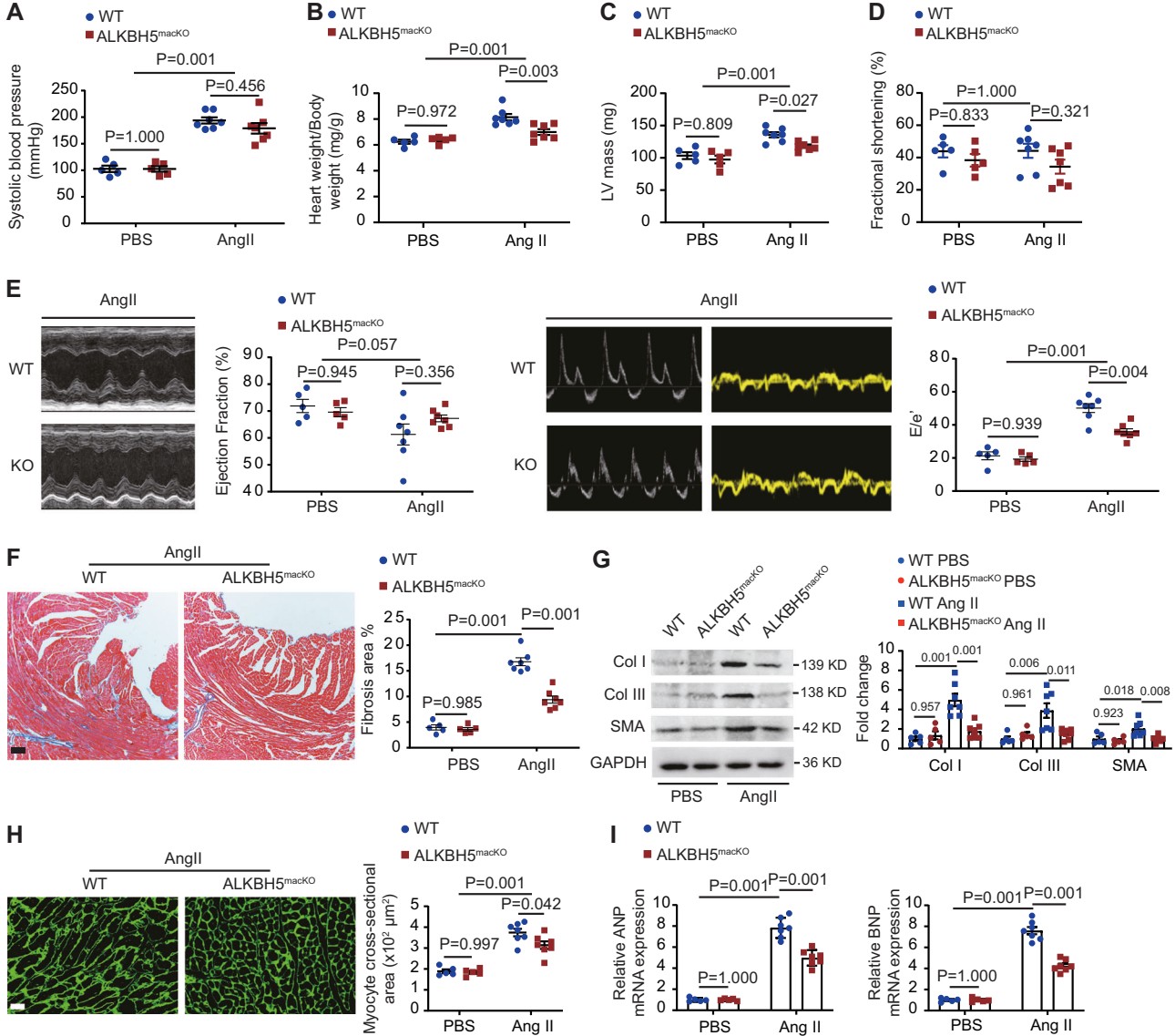

**Fig. 3 | Specific deletion of ALKBH5 in Cx3cr1 lineage improves Ang II induced MMT and cardiac fibrosis. A** Quantification analysis of SBP in sham and Ang II treated *WT* and *ALKBH5^(macKO-Td)* mice (*n* = 5 for PBS group and *n* = 7 for Ang II group). **B** Quantification analysis of heart weight/body weight in sham and Ang II treated *WT* and *ALKBH5^(macKO-Td)* mice (*n* = 5 for PBS group and *n* = 7 for Ang II group). **C** Quantification analysis of LV mass in sham and Ang II treated *WT* and *ALKBH5^(macKO-Td)* mice (*n* = 5 for PBS group and *n* = 7 for Ang II group). **D** Quantification analysis of fractional shortening in sham and Ang II treated *WT* and *ALKBH5^(macKO-Td)* mice (*n* = 5 for PBS group and *n* = 7 for Ang II group). **E** Representative echocardiography images and quantification of ejection fraction (left) and E/e' (right) of *WT* and *ALKBH5^(macKO-Td)* mice after PBS or Ang II treatment for 14 days (*n* = 5 for PBS group and *n* = 7 for Ang II group). **F** Representative images of Masson trichrome staining in cardiac tissue and quantification of positive fibrotic area (*n* = 5 for PBS group and *n* = 7 for Ang II

group). Scale bar, 100 μm. **G** Representative images and quantification of the SMA and collagen (Col) types I and III expression by western blot in cardiac tissues of *WT* and *ALKBH5^(macKO-Td)* mice (*n* = 5 for PBS group and *n* = 7 for Ang II group). **H** Representative images of wheat germ agglutinin staining in cardiac tissue and quantification of positive fibrotic area (*n* = 5 for PBS group and *n* = 7 for Ang II group). Scale bar, 100 μm. **I** mRNA levels of ANP and BNP expression in cardiac Td+ cells from *WT* and *ALKBH5^(macKO)* mice. *n* = 5 for PBS group and *n* = 7 for Ang II group. All data are presented as mean ± standard error mean. Data in **A**–**F** and **H** were analyzed by two-way ANOVA followed by Tukey post-hoc tests. Data in **G**, **I** were analyzed by one-way ANOVA followed by Tukey post-hoc tests. n.s. indicates nonsignificant. *P* < 0.05 was considered as statistically significant. Ang II Angiotensin II, WT wild-type, ANP atrialnatriureticpeptide, BNP brain natriuretic peptide, Col collagen.

exhibited a significant decrease of heart weight/body weight (Fig. 3B), LV mass (Fig. 3C) and E/e' ratio compared with control mice (Fig. 3F). Cardiac fibrosis evaluated by Masson's Trichrome-staining (Fig. 3F), as well as the expression of SMA and ECM genes collagen I and III (Fig. 3G), were also decreased in the heart of *ALKBH5^(macKO-Td)* mice compared with control mice. *ALKBH5^(macKO-Td)* mice also exhibited smaller cardiomyocytes shown by wheat germ agglutinin staining (Fig. 3H), as well as attenuation of cardiac fetal gene atrial natriuretic peptide (ANP) and brain natriuretic peptide (BNP) reactivation (Fig. 3I).

These data suggest that loss of ALKBH5 in macrophage inhibits MMT and improves pathological cardiac fibrosis and hypertrophy and further attenuates cardiac diastolic dysfunction.

Macrophage proliferation and cardiac fibroblast activation play pivotal roles in pathological cardiac remodeling[20]. Therefore, we evaluated the effects of ALKBH5 on Cx3cr1-derived cell proliferation, and found that the proliferation (Ki67+ cells) of Cx3cr1-derived Td positive cells in the heart of *ALKBH5^(macKO)* mice was decreased (Supplementary Fig. 5B). Consistently, cell proliferation of cultured

macrophages was decreased after ALKBH5 knockout with Ang II-stimulation (Supplementary Fig. 5C). We also performed macrophage and cardiac fibroblast co-culture assay (Supplementary Fig. 5D), and found that ALKBH5 deletion in macrophages significantly inhibited SMA and ECM genes collagen I and III expression in Ang II-treated cardiac fibroblast (Supplementary Fig. 5E).

To further validate the roles of ALKBH5 in pathological cardiac remodeling, we analyzed pressure overload-induced cardiac dysfunction by utilizing transverse aortic constriction (TAC) mouse model, and found a greater decrease of E/e' ratio and increase of ejection fraction in *ALKBH5^{macKO-Td}* mice after TAC 28 days (Supplementary Fig. 6A). Further, *ALKBH5^{macKO-Td}* mice displayed decreased SMA+Td+ cells and cardiac fibrosis shown by Masson's Trichrome-staining and the expression of SMA and ECM genes collagen I and III (Supplementary Fig. 6B–D). These data indicate that ALKBH5 deletion in Cx3cr1 lineage improves cardiac diastolic function under hypertensive stress.

### ALKBH5 in circulating monocytes-derived macrophage contributes to hypertension-induced cardiac fibrosis and dysfunction

To determine the effects of ALKBH5 in cardiac resident macrophages on MMT and cardiac fibrosis, we generated *Cx3cr1Cre^{ERT2}; Rosa26^{Td}; ALKBH5^{flox/flox}* mice to delete ALKBH5 in cardiac resident macrophages (Supplementary Fig. 7A). As the previous study[12], Td+ mainly labeled the cardiac Lyve1+ resident macrophages, and ALKBH5 deletion had no significant effect on Td+ cell cluster under both steady and hypertensive state (Supplementary Fig. 7B). Compared with the *Cx3cr1Cre^{ERT2}; Rosa26^{Td}; ALKBH5^{wt/wt}* (*Cx3cr1Cre^{ERT2}; Rosa26^{Td}*) mice, *Cx3cr1Cre^{ERT2}; Rosa26^{Td}; ALKBH5^{flox/flox}* mice showed no significant change in hypertension-induced MMT assessed by FACS in hearts (Supplementary Fig. 7B). No difference of cardiac fibrosis and diastolic function were observed in *Cx3cr1Cre^{ERT2}; Rosa26^{Td}; ALKBH5^{flox/flox}* mice compared with *Cx3cr1Cre; Rosa26^{Td}* mice under both steady and hypertensive state (Supplementary Fig. 7C–E).

Since the marker gene expression could not absolutely distinguishes circulating monocyte-derived macrophages from cardiac resident macrophages, we performed parabiosis assay by conjoining *CCR2^{KO}* mice, which lacked peripheral blood Ly6c^{hi} monocytes (Fig. S8A, B), with *Cx3cr1Cre; Rosa26^{Td}; ALKBH5^{wt/wt}* or *Cx3cr1Cre; ALKBH5^{fl/fl}; Rosa26^{Td}* mice, respectively (Fig. 4A). There was no difference of chimeric Td+ cells in hearts and blood from *CCR2^{KO}* mice conjoined with *Cx3cr1Cre; Rosa26^{Td}; ALKBH5^{wt/wt}* mice (Td-(WT)) and *Cx3cr1Cre; ALKBH5^{fl/fl}; Rosa26^{Td}* mice (Td-(KO)) (Fig. 4B, Supplementary Fig. 8C). Importantly, FACS analysis showed that *CCR2^{KO}* mice conjoined with *Cx3cr1Cre; ALKBH5^{fl/fl}; Rosa26^{Td}* mice displayed decreased percentage of CD11b^+SMA^+ cells in Td^+ cells compared to *Cx3cr1Cre; Rosa26^{Td}; ALKBH5^{wt/wt}* mice after AngII-infusion (Fig. 4B), which were further confirmed by immunostaining of SMA in hearts of conjoined *CCR2^{KO}* mice (Fig. 4C). These data suggested that loss of ALKBH5 attenuates the infiltration of circulating monocytes and subsequently inhibits MMT of these monocyte-derived macrophages.

*CCR2^{KO}* mice received blood from *Cx3cr1Cre; ALKBH5^{fl/fl}; Rosa26^{Td}* mice showed decreased E/e' ratio than *Cx3cr1Cre; Rosa26^{Td}; ALKBH5^{wt/wt}* mice (Fig. 4D, E). Masson trichrome staining showed a significant reduction of cardiac fibrosis in *CCR2^{KO}* mice conjoined with *Cx3cr1Cre; ALKBH5^{fl/fl}; Rosa26^{Td}* compared with mice conjoined with *Cx3cr1Cre; Rosa26^{Td}; ALKBH5^{wt/wt}* mice (Fig. 4F, G), indicating that ALKBH5 in circulating monocytes/macrophages contributed to hypertension-induced cardiac fibrosis and dysfunction. Decreased SMA and ECM genes collagen I and III in the heart were observed in *CCR2^{KO}* mice conjoined with *Cx3cr1Cre; ALKBH5^{fl/fl}; Rosa26^{Td}* compared with *CCR2^{KO}* mice conjoined with *Cx3cr1Cre; Rosa26^{Td}; ALKBH5^{wt/wt}* mice (Fig. 4H).

### ALKBH5 promotes MMT and cardiac fibrosis via regulating m6A modification on IL-11 mRNA in Cx3cr1-derived macrophages

To investigate the mechanism by which ALKBH5 regulates MMT and cardiac fibrosis under hypertension, we performed ALKBH5 RNA immunoprecipitation-sequencing (RIP-seq) in cultured macrophages and found several genes are the direct targets of ALKBH5 (Fig. 5A). The pathway enrichment analysis was performed to show the pathway regulated by ALKBH5 responsive to Ang II. Among the top 10 enriched pathways, we observed that Tgf-beita signaling pathway was associated with ALKBH5, which further indicated that macrophage ALKBH5 participated in the process of Ang II-induced macrophage-myofibroblast transition (Fig. 5B). Moreover, Ang II significantly increased the enrichment of ALKBH5 on IL-11 mRNA, the top abundant enriched target mRNA (Fig. 5A, Supplementary Data 1). Therefore, we hypothesized that IL-11 mRNA might be the m6A target directly modulated by ALKBH5. We next did m6A RNA immunoprecipitation-sequencing in ALKBH5 knockout and control macrophages and found that the GGAC motif was highly enriched within m⁶A locus in both control and ALKBH5 knockout macrophages (Fig. 5C). The m⁶A peaks were particularly abundant near start and stop codons (Fig. 5D). m⁶A peaks were enriched on IL-11 mRNA (Fig. 5E), which was confirmed by m6A RIP-qPCR using primer on peak1 and peak2 regions (Fig. 5F). We then assessed RNA decay modulated by m6A and found that IL-11 mRNA decreased faster in ALKBH5 deficient macrophages (Fig. 5G). These data suggest that ALKBH5 deficiency in macrophages increases m6A modification on IL-11 mRNA, which results in decreased IL-11 stability.

Previous scRNA-seq analysis of fibrotic hearts demonstrated that IL-11 and its receptor (IL11RA1) pathway is a crucial determinant for cardiovascular fibrosis[21]. Our scRNA sequencing data also showed that macrophage-derived myofibroblast cluster, but not macrophage, was highly enriched for IL11RA1 under hypertensive stress (Fig. 5H). Pseudotime analysis also showed that IL11RA1 expression level was in parallel with the process of MMT in cardiac macrophages (Fig. 5I). The expression of IL11 and IL11RA1 were increased in sorted Td+ cells from *Cx3cr1Cre;Rosa26^{Td}* following Ang II-infusion (Fig. 5J). These suggested that ALKBH5 promotes cardiac macrophage-myofibroblast transition and fibrosis via directly targeting IL-11 mRNA and subsequently activating IL-11/IL11RA1 pathway in cardiac macrophages.

To identify the role of IL-11 in ALKBH5-mediated cardiac dysfunction, we utilized bone-marrow transplantation (BMT) to specifically overexpress IL11 in circulating monocytes. FACS analysis indicated the success of BMT, with Td+ cells in the reconstituted hearts of both *ALKBH5^{macKO-Td}-BM* and *Cx3cr1Cre^{Td}-BM* mice (Fig. 6A). ALKBH5 deficiency or IL11 overexpression did not influence cardiac Td+ cells infiltration (Fig. 6A). FACS analysis and immunostaining analysis showed that C57BL6/J mice transplanted with BM from *ALKBH5^{macKO}-Td* mice displayed decreased percentage of SMA^+ in Td^+ cells compared to *Cx3cr1Cre^{Td}* mice after AngII-infusion, which were elevated after IL11 overexpression (Fig. 6A, B). Echocardiography data showed that IL11 overexpression (IL11 OE) damaged the cardiac diastolic function shown by the increase in E/e' while no change in ejection fraction in both *ALKBH5^{macKO-Td}-BM* and *Cx3cr1Cre^{Td}-BM* mice (Fig. 6C, D). IL11 overexpression reversed the decreased cardiac fibrosis (Fig. 6E) and fibrosis related genes caused by ALKBH5 knockout in macrophages (Fig. 6F).

Exogenous IL-11 recombinant protein (rIL-11) was also used in vitro to confirm the role of ALKBH5-IL11 pathway in Ang II induced MMT and subsequently proliferation of myofibroblasts. We found that rIL-11 promoted Ang II-induced SMA, Col I and Col III expression and cell proliferation in both wild type (WT) and ALKBH5 knockout (KO) macrophages (Supplementary Fig. 9A–C). Consistently, Co-cultured macrophages with primary cardiac fibroblasts demonstrated that rIL-11 administration blocked ALKBH5 knockout-mediated inhibition of SMA and ECM gene expression and proliferation of cardiac fibroblasts

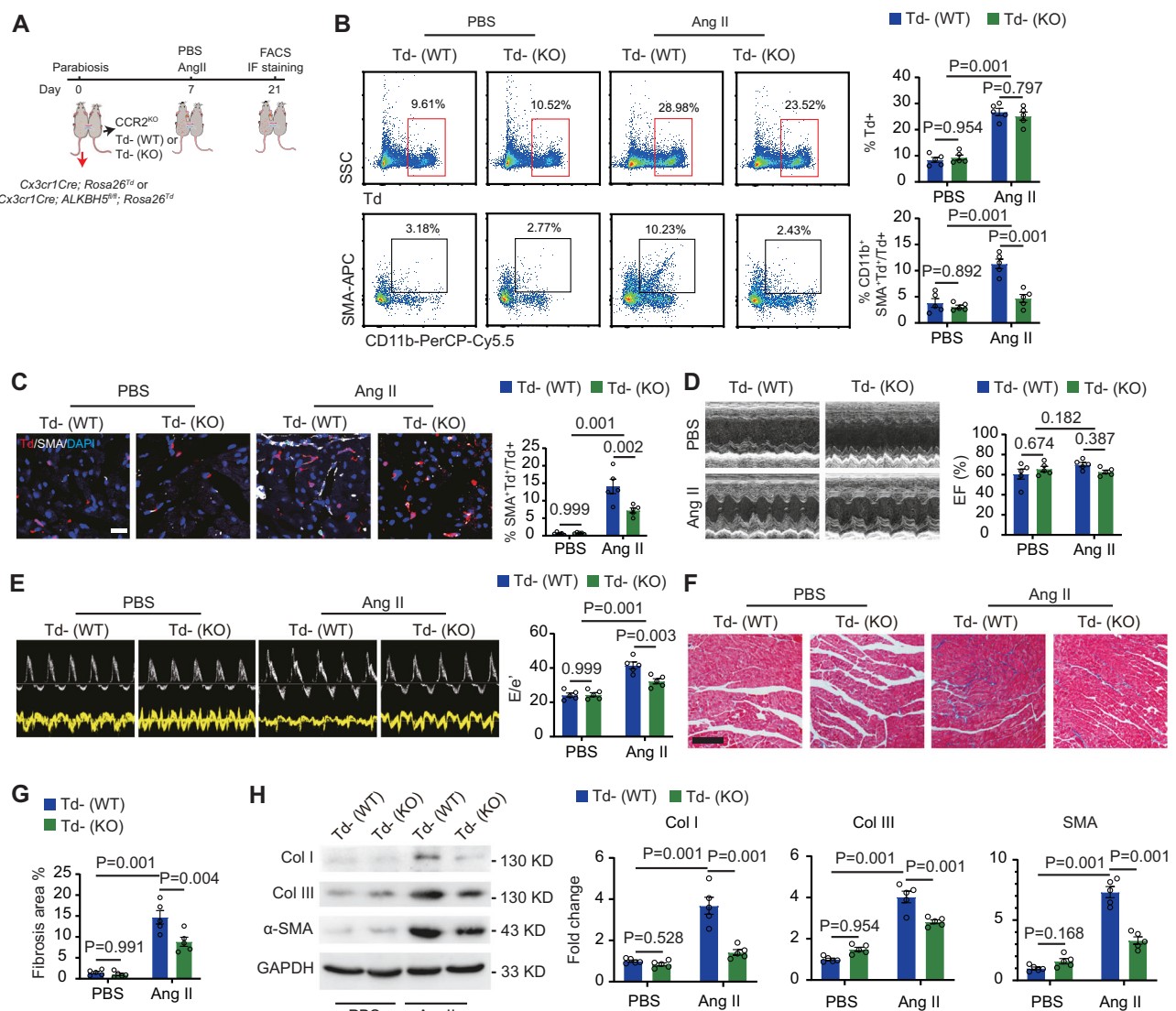

**Fig. 4 | ALKBH5 in circulating monocytes-derived macrophage contributes to hypertension-induced cardiac fibrosis and dysfunction. A** Diagram of parabiosis between *CCR2^KO* and *Cx3cr1Cre; Rosa26^Td* or *Cx3cr1Cre; ALKBH5^fl/fl; Rosa26^Td* mice, respectively. **B** Representative images and quantification of flow cytometry analyses of Td+ and CD11b+SMA+ cells gated on Td+ cells in hearts from *CCR2^KO* mice cojoined with *Cx3cr1Cre; Rosa26^Td* or *Cx3cr1Cre; ALKBH5^fl/fl; Rosa26^Td* mice. *n* = 5. **C** Representative immunofluorescent images and quantification of SMA+ cells in Td+ cells of cardiac tissues from *CCR2^KO* cojoined with *Cx3cr1Cre; Rosa26^Td* or *Cx3cr1Cre; ALKBH5^fl/fl; Rosa26^Td* mice (*n* = 5). Scale bar, 100 μm. **D** Representative echocardiography images of ejection fraction of the *CCR2^KO* cojoined with *Cx3cr1Cre; Rosa26^Td* or *Cx3cr1Cre; ALKBH5^fl/fl; Rosa26^Td* mice after Ang II treatment for 14 days, with indices of cardiac ejection fraction and E/e′ at right. *n* = 5.

**E** Representative echocardiography images of E/e′ (E) of the *CCR2^KO* cojoined with *Cx3cr1Cre; Rosa26^Td* or *Cx3cr1Cre; ALKBH5^fl/fl; Rosa26^Td* mice after Ang II treatment for 14 days, with indices of cardiac ejection fraction and E/e′ at right. *n* = 5. **F** Representative images of Masson trichrome staining. *n* = 5. Scale bar, 100 μm. **G** Quantification of positive fibrotic area. *n* = 5. Scale bar, 100 μm. **H** Representative images of SMA and ECM genes collagen I and III in cardiac tissues shown by western blot (*n* = 5). All data are presented as mean ± standard error mean. Data in **B**–**E** and **G** were analyzed by two-way ANOVA followed by Tukey post-hoc tests. Data in **H** were analyzed by one-way ANOVA followed by Tukey post-hoc tests. n.s. indicates nonsignificant. *P* < 0.05 was considered as statistically significant. Ang II Angiotensin II, WT wild-type, KO knockout, Col collagen.

(Supplementary Fig. 9D–F). To further investigate the role of IL11, we inhibited IL11 signaling in cultured macrophages using neutralizing antibody against IL11. The results showed that inhibition of IL11 signaling reduced SMA and Col1 expression and cell proliferation in WT macrophages, but had no effect on ALKBH5 knockout macrophages (Supplementary Fig. 9G, I). We also assessed the expression of ALKBH5, IL-11 and IL11RA1 in the cultured fibroblasts, and found that Ang II treatment had no significant effect on ALKBH5 and IL-11 expression, but increased IL11RA1 expression in cardiac fibroblasts (Supplementary Fig. 10). Overall, these in vivo and in vitro data inferred that ALKBH5/IL-11 pathway activation is mainly existed in cardiac macrophages. Then the secreted IL-11 induces MMT, as well as

promotes cardiac fibroblast activation via directly binding to the receptor IL11RA1 in both macrophages and fibroblasts.

## Nanoparticle monocyte/macrophage-target delivery of ALKBH5 or IL11RA1 siRNA attenuates Ang II-induced cardiac fibrosis and dysfunction

Lipid nanoparticle (LNP)-mRNA technology is widely used for targeting immune cells, especially for monocyte-derived macrophages in various animal models[22]. We therefore generated the DiR-modified LNPs C12-200[23,24] that is successfully used for targeting circulating monocytes (Fig. 7A). Size distribution and zeta potential of LNPs were measured by dynamic light scattering and cryo-transmission electron

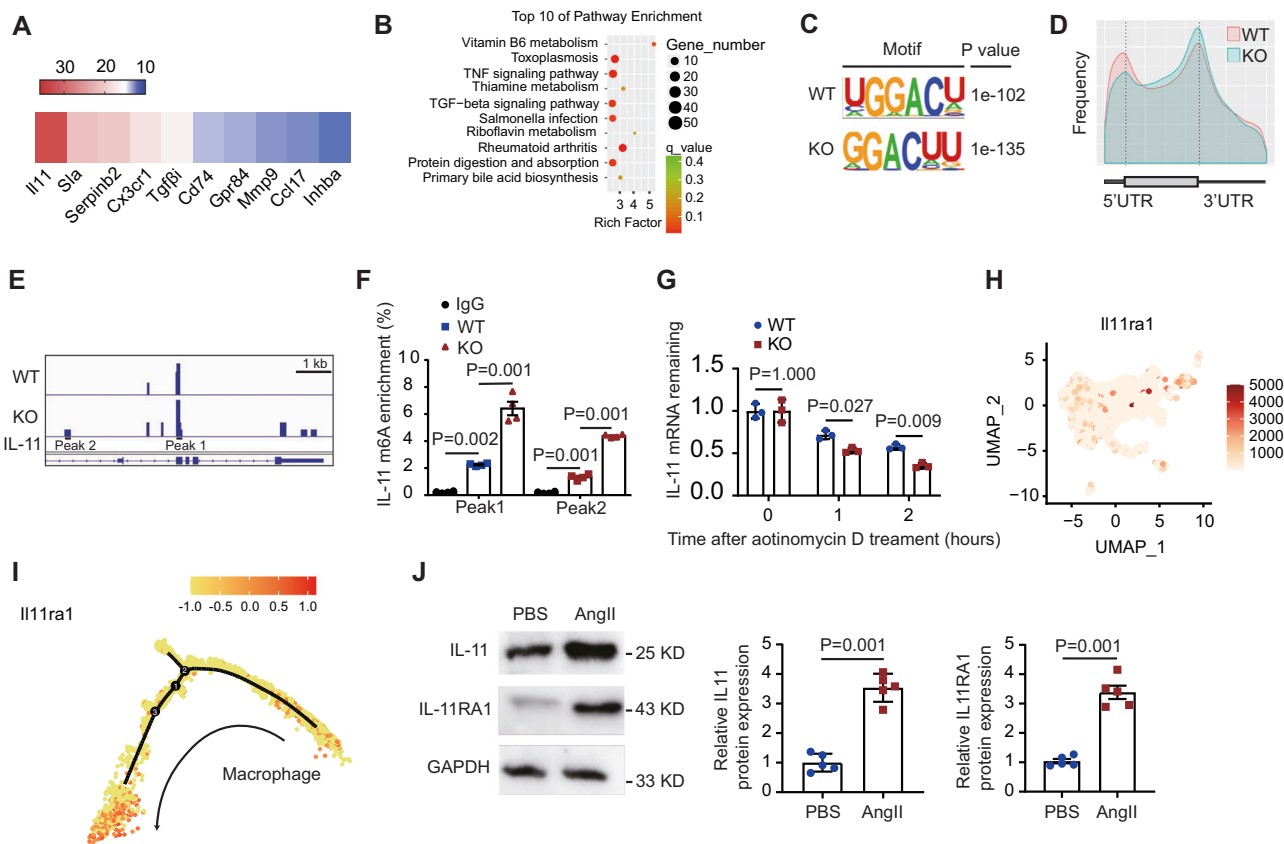

**Fig. 5 | Macrophage ALKBH5 directly regulates m6A modification on IL-11 mRNA. A** top abundant enriched target mRNAs of ALKBH5 induced by Angiotensin II. **B** Top enriched pathway of ALKBH5 induced by Angiotensin II. **C** m6A motif identified from m6A RNA immunoprecipitation-sequencing analysis. **D** Metagene distribution of m6A peaks along the whole transcriptome, including 5′ untranslated regions (5′ UTR), covering coding sequence (CDS) and 3′ UTR. **E** m6A peak distribution on IL-11 mRNA shown by Integrative Genomics Viewer (IGV) tool. kb means kilo base pair. **F** Relative fold change of m6A enrichment on IL-11 mRNA by m6A RIP-qPCR analysis ($n$ = 4). **G** Degradation of IL-11 mRNA was detected in macrophages following treating with actinomycin D for indicated time ($n$ = 3).

**H** IL11RA1 gene signatures visualized by Feature Plots. **I** Expression level of IL11RA1 along the pseudotime trajectory. **J** Protein levels of IL11RA1 in cardiac Td$^+$ cells of the hearts following PBS and Ang II treatment shown by western blot with representative images at left and quantification at right. $n$ = 5. All data are presented as mean ± standard error mean. Data in **F** were analyzed by one-way ANOVA followed by Tukey post-hoc tests. Data in **G** were analyzed by two-way ANOVA followed by Tukey post-hoc tests. Data in **J** were analyzed by two-tailed unpaired Student's $t$-test. n.s. indicates nonsignificant. $P$ < 0.05 was considered as statistically significant. UTR Untranslated Regions, WT wild-type, KO knockout, UMAP uniform manifold approximation and projection, Ang II angiotensin II.

micrographs (Fig. 7B–D). We then delivered ALKBH5 or IL11RA1 siRNA encapsulated in monocyte/macrophage-target DiR-modified LNPs C12-200 to alleviate the cardiac fibrosis and dysfunction under hypertensive stress (Fig. 7E). Fluorescence imaging combined with microCT showed that nanoparticles was successfully distributed in the heart following Ang II-infusion (Fig. 7F). ALKBH5 siRNA-encapsulated LNPs improved Ang II-induced cardiac dysfunction shown by decreased E/e′ ratio (Fig. 7G, H). Masson trichrome staining showed that LNP-mediated ALKBH5 silencing significantly improved cardiac fibrosis caused by Ang II-infusion (Fig. 7I). ALKBH5 silencing decreased SMA positive cells (Fig. 7J), and also expression of SMA and collagen types I and III in the Ang II-infused hearts (Fig. 7K). Similarly, IL11RA1 siRNA-encapsulated LNPs treatment improved cardiac function and attenuated cardiac fibrosis (Supplementary Fig. 11A–D). These suggest that targeting macrophage-ALKBH5-IL-11/IL11RA1 pathway may provide a novel therapy for pathological cardiac remodeling.

## Discussion

Macrophages play crucial roles in pathological cardiac fibrosis. However, the factors that regulate the phenotype or transition and activation of cardiac macrophages during cardiac fibrosis is not clear. Herein, we provide evidence that circulating monocyte-derived cardiac macrophages transition to myofibroblasts, and contribute to

cardiac fibrosis under hypertension. ALKBH5 promotes MMT and activates cardiac fibroblasts through upregulating IL-11 in Cx3cr1 precursor-derived cells. Macrophage-targeted lipid nanoparticles, containing ALKBH5 or IL11RA1 siRNA, attenuate the transition of macrophages to myofibroblasts and improve hypertensive cardiac fibrosis (Fig. 8). Taken together, the present study provides direct evidence that transition of macrophages to myofibroblasts is involved in the process of hypertension-induced cardiac fibrosis, which is largely dependent on the ALKBH5-decreased m6A-demethylation on IL-11 in cardiac macrophages.

Transition of macrophages has been reported in different pathological processes, probably due to the plasticity of the myeloid lineage, especially monocytes/macrophages. Increasing evidence suggests that the historically defined fibroblast is actually not a cell type, but a general name for heterogeneous cells. A number of cell types, including BM-derived cells and endothelial cells, are suggested to contribute to pathological fibrosis by converting to a myofibroblast phenotype via utilizing BM chimeras, parabiosis, and lineage tracing experimental approaches[25]. A majority of cardiac fibroblasts is derived from epicardial and endothelial cells that express Tcf21, Wt1 or Tbx18 during embryonic development[26–28]. In addition to this, bone marrow-derived myeloid cells, fibrocytes, and small infiltrating immune cells have been suggested to generate

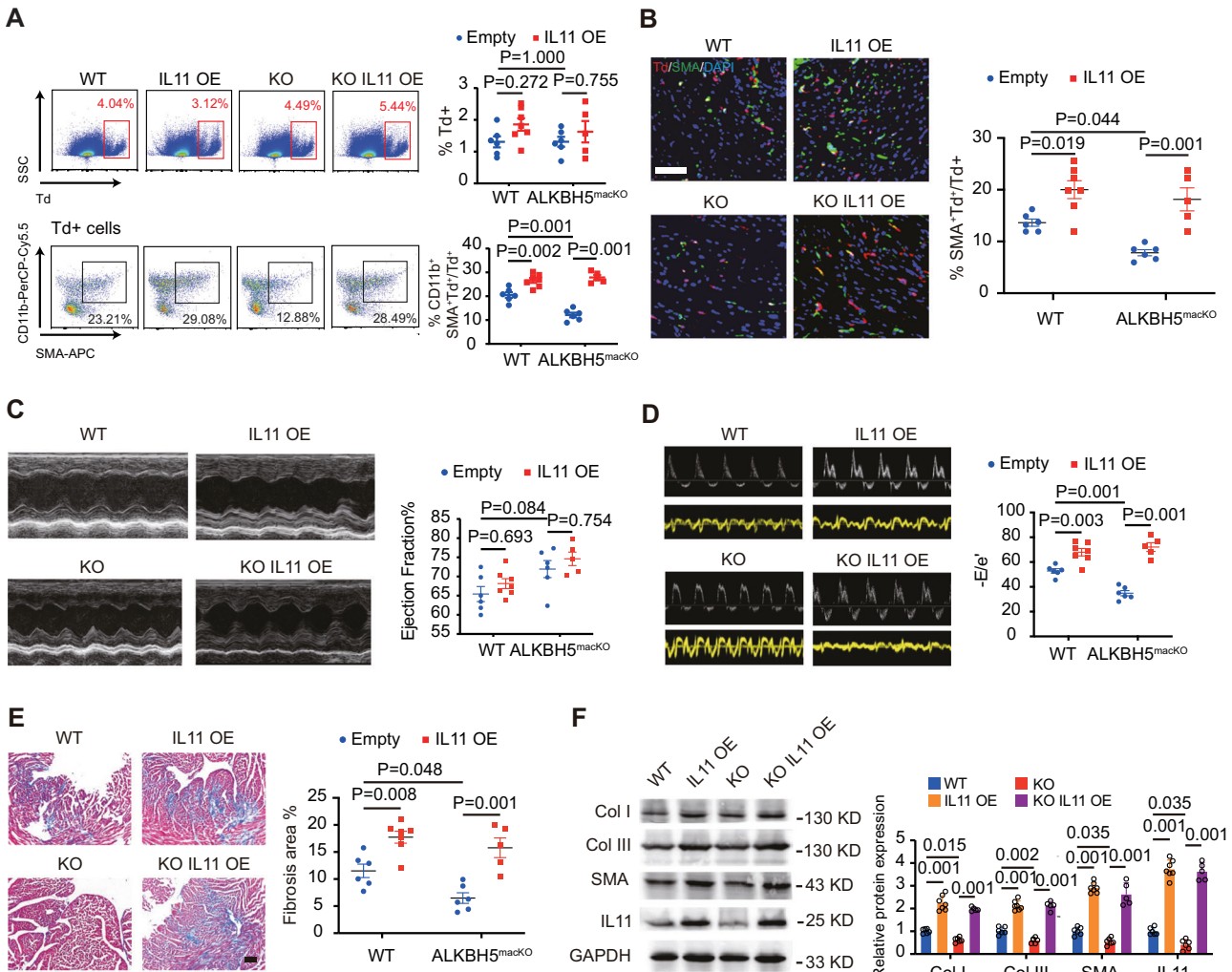

**Fig. 6 | IL-11 overexpression reverses macrophage ALKBH5 deletion-mediated improved cardiac dysfunction via increasing cardiac fibrosis. A** Representative flow cytometry analyses of Td+ and CD11b+SMA+cells in hearts from C57BL6/J mice transplanted bone marrow from *ALKBH5macKO-Td-BM* and *Cx3cr1CreTd-BM* mice with empty vector or IL11 overexpression, with quantification at right. *n* = 6 for *WT* and *ALKBH5macKO* group. *n* = 7 for IL11 OE group. *n* = 5 for *ALKBH5macKO* and IL11 OE group. **B** Representative immunofluorescent images and quantification of SMA⁺ cells in Td⁺ cells of cardiac tissues from the reconstituted mice. *n* = 6 for *WT* and *ALKBH5macKO* group. *n* = 7 for IL11 OE group. *n* = 5 for *ALKBH5macKO* and IL11 OE group. Scale bar, 100 μm. **C** Representative echocardiography images and quantification of ejection fraction of the above mice after Ang II treatment for 2 weeks. *n* = 6 for *WT* and *ALKBH5macKO* group. *n* = 7 for IL11 OE group. *n* = 5 for *ALKBH5macKO* and IL11 OE group. **D** Representative echocardiography images and

quantification of E/e' of the above mice after Ang II treatment for 2 weeks. *n* = 6 for *WT* and *ALKBH5macKO* group. *n* = 7 for IL11 OE group. *n* = 5 for *ALKBH5macKO* and IL11 OE group. **E** Representative images of Masson trichrome staining in cardiac tissue and quantification of positive fibrotic area. *n* = 6 for *WT* and *ALKBH5macKO* group. *n* = 7 for IL11 OE group. *n* = 5 for *ALKBH5macKO* and IL11 OE group. Scale bar, 100 μm. **F** Representative images and quantification of the SMA, collagen (Col) types I and III and IL11 expression by western blot in cardiac tissues of the reconstituted mice. *n* = 6 for *WT* and *ALKBH5macKO* group. *n* = 7 for IL11 OE group. *n* = 5 for *ALKBH5macKO* and IL11 OE group. All data are presented as mean ± standard error mean. Data in **A**−**E** and **F** were analyzed by one-way ANOVA followed by Tukey post-hoc tests. n.s. indicates nonsignificant. *P* < 0.05 was considered as statistically significant. WT wild-type, KO knockout, OE overexpression, Col collagen.

myofibroblasts in the diseased mouse heart[29–31]. There are two major populations of macrophages within the healthy and diseased heart, a population of CCR2⁺ circulating monocyte-derived macrophages and a smaller population of resident cardiac Lyve1⁺ macrophages, that could be replaced by infiltrating monocyte-derived CCR2⁺ macrophages[32,33]. In unstressed hearts of mice, the embryonic Cx3cr1 precursor-derived resident cardiac macrophages facilitate tissue homeostasis with expressing Lyve1[10,34]. Cx3cr1 precursors also labels circulating monocytes and that infiltrates to diseased heart to generate CCR2⁺ macrophages, that causes an acute sterile immune response during cardiac cell therapy remodeling[12,13,32,35]. Our genetic fate mapping supported that circulating monocyte-derived cardiac macrophages transition to myofibroblasts, which contributes to cardiac fibrosis under hypertension.

N6-methyladenosine (m6A), the most abundant modification on mRNA, is catalyzed by methyltransferase complex and demethylases. m6A modification controls various aspects of immunity, including immune recognition, activation of innate and adaptive immune responses, and immune cell development and fate decisions[36]. ALKBH5-mediated m6A-demethylation has been reported to be involved in the regulation of biological processes, including immune response, ROS production and viral infection[37–39]. Ablation of ALKBH5 in CD4⁺ T cells increases m6A modification on IFN-γ and CXCL2 mRNA and reduces corresponding protein expression, thereby attenuating T cell-mediated inflammation and immune response[40]. In the cardiovascular system, ALKBH5 plays different roles in different cell types. ALKBH5 promotes cardiomyocyte proliferation after myocardial infarction while attenuating endothelial cell-mediated post-ischemic

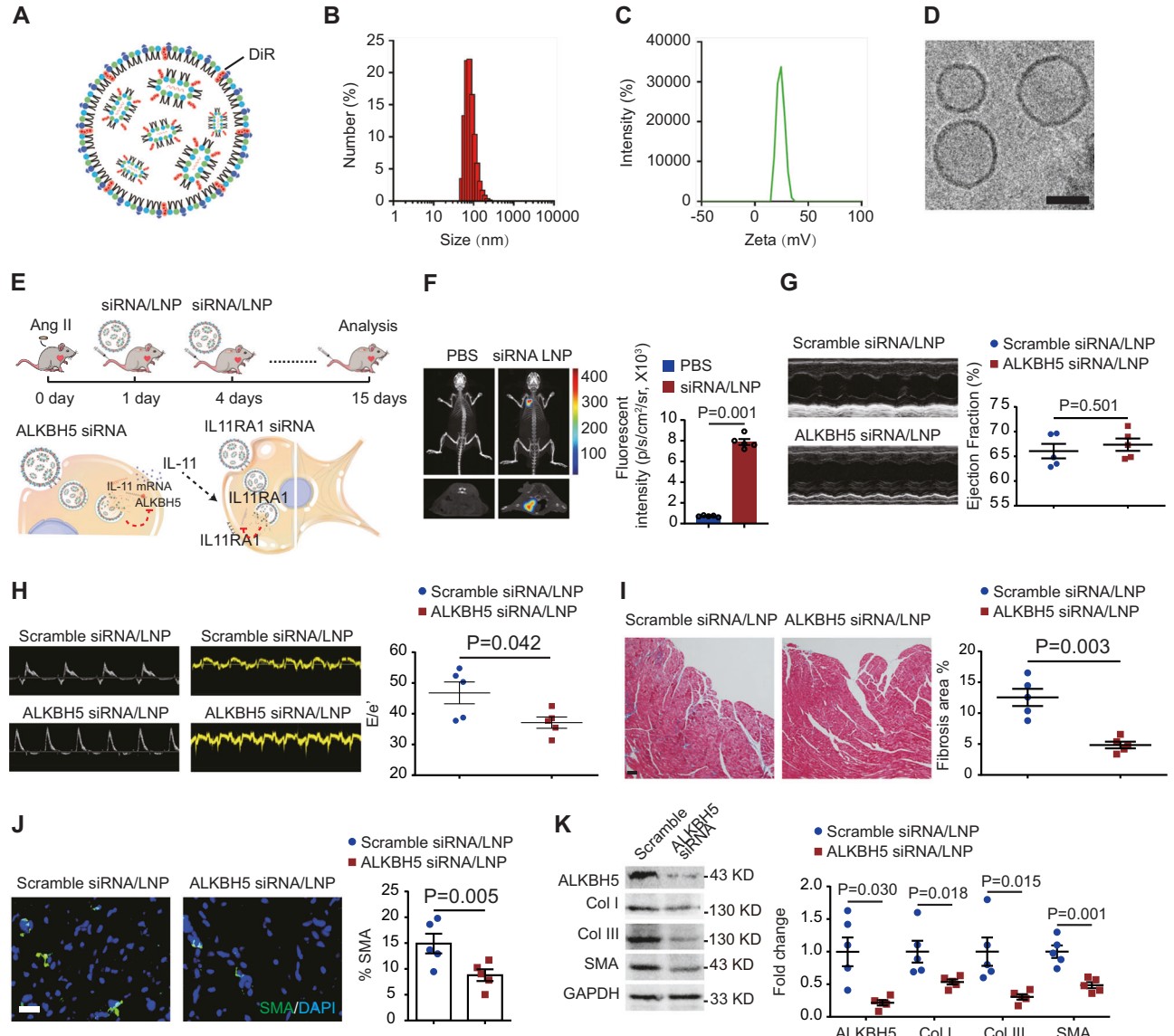

**Fig. 7 | Nanoparticle monocyte/macrophage-target delivery of ALKBH5 siRNA attenuates Ang II-induced cardiac fibrosis and dysfunction. A** Schematic diagram of lipid nanoparticle. **B** Size distribution of lipid nanoparticle measured by dynamic light scattering (DLS). **C** Zeta potential of lipid nanoparticle measured by dynamic light scattering (DLS). **D** The represent cryo-transmission electron micrographs (TEMS) of lipid nanoparticle. Scale bar, 50 nm. **E** Experimental diagram of in vivo ALKBH5 loss-of-function using C12-200 lipid nanoparticles. **F** Representative fluorescence imaging combined with microCT after intravenous injection of DiR-labeled C12-200 lipid nanoparticles, with quantification of fluorescent intensity at right. $n = 5$. **G** Representative echocardiography images and indices of ejection fraction of mice with scramble or ALKBH5 siRNA/LNP. $n = 5$. **H** Representative echocardiography images and indices of E/e' of mice with

scramble or ALKBH5 siRNA/LNP. $n = 5$. **I** Representative images (left) and quantification of positive fibrotic area (right) of Masson trichrome staining in cardiac tissue of mice with scramble or ALKBH5 siRNA/LNP $n = 5$. Scale bar, 100 μm. **J** Representative immunofluorescent images and quantification of SMA+ cells in cardiac tissues from scramble or ALKBH5 siRNA/LNP treated mice. $n = 5$. Scale bar, 100 μm. **K** Western blot analysis of IL11RA and ECM genes collagen I and III in cardiac tissues from mice with scramble or ALKBH5 siRNA/LNP. Quantitative results are shown on the right. $n = 5$. All data are presented as mean ± standard error mean. Data in **F**−**K** were analyzed by two-tailed unpaired Student's $t$-test. n.s. indicates nonsignificant. $P < 0.05$ was considered as statistically significant. siRNA small interfering RNA, LNP lipid nanoparticle, Col collagen.

angiogenesis[40,41]. Herein, we provide the latest advance that ALKBH5 regulates cardiac macrophage transition and function via direct modification of IL-11 mRNA during pathological cardiac fibrosis. IL-11 are previously reported to be expressed mainly in a small population of cardiac myofibroblasts according to scRNA sequencing data[21]. IL-11 drives fibrogenic protein synthesis and cardiac fibrosis via activating its receptor IL11RA1, specially expressed in fibroblasts. We found that IL11RA1 are mainly expressed in myofibroblasts. TGFβ is reported to stimulate IL-11 expression in cardiac fibroblasts. Herein, we provide a novel post-transcriptional mechanism in regulating IL-11 level in Cx3cr1 precursor-derived myofibroblasts, that ALKBH5 decreases m6A

modification of IL-11 mRNA, and increases IL-11 mRNA stability and corresponding protein level. Loss of ALKBH5 in cardiac macrophages inhibits the transition of macrophages to myofibroblasts, and improves cardiac fibrosis after Ang II infusion. Whereas over-expression of IL11 in bone marrow-derived monocyte/macrophage blocks the protective effects against pathological cardiac fibrosis in *Cx3cr1cre; ALKBH5flox/flox* mice. Though our m6A sequencing profile showed weak signal change for IL-11 in ALKBH5 knockout macrophages, the fold change of IL-11 in ALKBH5 RIP-seq is at the top of the list. Therefore, we undertook comprehensive considerations of m6A sequencing and RIPP-seq, and found that IL11 was the critical candidate

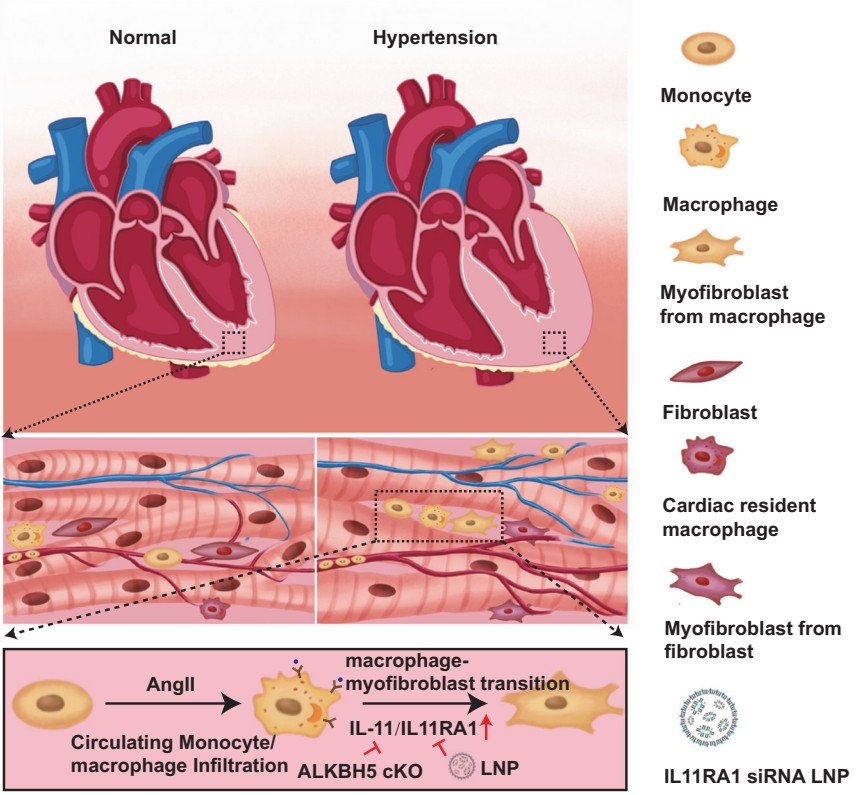

**Fig. 8 | Schematic illustration of macrophage ALKBH5-IL-11/IL11RA1 pathway in cardiac remodeling.** Schematic illustration of macrophage ALKBH5-IL-11/IL11RA1 pathway in cardiac remodeling.

gene for the phenotype in this mode. However, we could not exclude the other ALKBH5 downstream candidate genes that may be involved in the regulation of this phenotype in the macrophages. This needs further investigation in the future.

Lipid nanoparticle-mRNA technology underlies recent successes in COVID-19 vaccine development and has stimulated broad efforts to extend this therapeutic platform to address numerous pathologies. A recent study reported an immunotherapy strategy to generate transient CAR T cells that can recognize the fibrotic cells in the heart by injecting CD5-targeted LNPs[42]. In fact, cardiac monocytes and macrophage abundance are much more than the other immune cells in the cardiac pathophysiological status. Therefore, we utilized optimized LNPs to target monocytes/macrophages, and ALKBH5 or IL11RA1-silencing siRNA is efficiently delivered to cardiac macrophages. This LNP-siRNA technology successfully inhibits the transition of macrophages to myofibroblasts, and improves pathological cardiac fibrosis. Taken together, our finding of ALKBH5-IL-11-IL11RA1 pathway activation-mediated transition of macrophage to myofibroblasts represents a novel mechanism that contributes to pathological cardiac fibrosis. Notwithstanding the much more advanced complexity of human pathological cardiac fibrosis in comparison with the mouse model, our novel findings suggest an attractive possibility that targeting ALKBH5-IL-11-IL11RA1 pathway in cardiac macrophages via LNP-siRNA technology might serve as a potential therapeutic tool for the prevention of cardiac fibrosis in heart diseases.

## Methods
### Animal
Male C57BL/6 mice were purchased from SLAC Laboratory Animal Co, Ltd (Shanghai, China). The conditional ALKBH5 mice (*ALKBH5^flox/+*) mice were purchased from Cyagen company (Stock#: S-CKO-09656).

*Cx3cr1Cre* mice[43] were obtained from the Jackson Laboratory (Stock#: 025524). *Lyve1Cre* mice[44] were from the Jackson Laboratory (Stock#: 012601). *Lyz2Cre*[45] (Stock#: 004781), *Cx3cr1Cre^ERT2*[46] (Stock#: 021160) and *Myh11Cre^ERT2*[47] (Stock#: 019079) mice were purchased from the Jackson Laboratory. *CCR2GFP* mice were from the Jackson Laboratory (Stock#: 027619). CCR2 knockout mice were obtained by mating *CCR2GFP/+* mice with each other. All mice were in C57BL/6J background and male mice aged 8-12 weeks were included in this study, and housed at 25 °C, 12-h light/dark in a specific pathogen-free environment at the University Laboratory Animal Services Centre of Fudan University.

### Fate-mapping
The *Cx3cr1Cre*, *Lyz2Cre*, *Cx3cr1Cre^ERT2*, *Lyve1Cre* and *Myh11Cre^ERT2* mice were crossed with *Rosa26^Td* (Stock#: 007914) to trace cardiac monocytes/macrophages. To label cardiac resident macrophages in *Cx3cr1Cre^ERT2;Rosa26^Td* mice, 3-4 week mice were given Tamoxifen (100 mg/kg, intraperitoneal every other day for total 4 times, ABCONE, T56488), and kept for 4 weeks after tamoxifen discontinuation to replenish the monocyte-dependent macrophages[12].

### Inactivation ALKBH5 in cardiac macrophages
*ALKBH5^flox/+* mice were crossed with *Cx3cr1Cre;Rosa26^Td* or *Cx3cr1Cre^ERT2;Rosa26^Td* mice to generate macrophage specific ALKBH5 knockout (*Cx3cr1Cre;Rosa26^Td;ALKBH5^flox/flox*) or Tamoxifen inducible knockout (*Cx3cr1Cre^ERT2; Rosa26^Td; ALKBH5^flox/flox*) mice.

3-4 week *Cx3cr1Cre;Rosa26^Td;ALKBH5^flox/flox* mice were administrated with Tamoxifen (100 mg/kg, intraperitoneal every other day for total 4 times, ABCONE, T56488), and kept for 4 weeks after tamoxifen discontinuation to inactivate ALKBH5 in cardiac resident macrophages.

## Angiotensin II infusion

Mice were anesthetized with 1% isoflurane and osmotic mini-pumps (Alzet) containing either saline or angiotensin II (1.44 mg/kg/day; Meilunbio, MB1677-2) were implanted under back for 2 weeks.

## SBP measurements

Systolic blood pressure (SBP) was measured by the noninvasive tail-cuff technique (Visitech Systems, Inc). Blood pressure were averaged from at least 10 consecutive measurements for each mouse.

## Transverse aortic constriction left ventricle pressure overload model

Briefly, under sterile conditions, a blunted 27-gauge needle placed parallel to the transverse aorta was tied by 7-0 silk suture. Subsequently, the needle was gently removed. The sham-operated mice were conducted with identical surgery except for tying the transverse aorta. Echocardiography was performed to evaluate the cardiac function four weeks later.

## Formulation of siRNA lipid nanoparticles (LNPs) for in vivo gene expression silencing

C12-200-siRNA liposomes comprised C12-200, DSPC, cholesterol, and mPEG2000-DMG. Briefly, the molar ratio of liposomes was C12-200/DSPC/cholesterol/mPEG2000-DMG at 50:10:38.5:1.5. The lipids were dissolved in 90% ethanol and the siRNA was solubilized in 10 mM citrate, pH3 buffer at a concentration of 0.4 mg/mL. Mixed lipids were slowly added to siRNA citrate buffer and incubated at room temperature for 30 min. Ethanol removal and buffer exchange of siRNA-containing liposomes was achieved by ultrafiltration against PBS using a 100,000 MWCO membrane. Finally, the formulation was filtered through a 0.2 mm sterile filter. Particle size was determined using a Malvern Zetasizer NanoZS (Malvern). The morphology of liposome was observed via cryo-TEM (FEI Tecnai G2 F20 electron microscope (200 kV)). The resulting particles had a mean particle diameter of 100 nm with approximately 90% siRNA entrapment efficiency. The liposomes were stained with DiR by directly adding DiR (1,1-dioctadecyl-3,3,3,3-tetramethylindotricarbocyanine iodide, 1 mol%) to the lipid mixture in ethanol during preparation.

## In vivo loss-of-function study by using siRNA-lipid nanoparticles (LNP) complexes

Briefly, siRNA lipid nanoparticles were diluted in PBS at a dose of 0.2 mg/ml. Formulations were administered intravenously via tail vein injection (2 mg/kg) for 14 days at an interval of 2 days. The distribution of siRNA lipid nanoparticles was captured with IVIS system. The efficiency of siRNA was determined by western blot.

## Dynamic FMT-CT imaging of siRNA biodistribution

Mice were placed onto the warmed stage inside the camera box and received continuous exposure to 2.5% isoflurane to sustain sedation during imaging. The light emitted from DiR dyes were detected, integrated, digitized, and displayed by the IVIS camera system. Regions of interest from displayed images were identified and quantified as total photon counts or photons/s using Living Image® software 4.0 (Caliper, Alameda, CA.).

## Echocardiography

Mouse echocardiography was performed using the Vevo2100 High-Resolution Imaging System. Two-dimensional long-axis and short-axis LV imaging were obtained for analysis. M-mode tracings were recorded through septum and posterior LV walls to measure LV dimension and wall thickness. Ejection fraction was calculated by measurement with LV end-diastolic diameter and end-systolic diameter. Pulse-wave Doppler and tissue Doppler imaging of apical 4-chamber view were recorded to measure and analyze E/e' (the ratio of peak early transmitral flow velocity to the peak early diastolic mitral annular velocity).

## Bone marrow transplantation (BMT)

Briefly, donor bone marrow (BM) was isolated from $ALKBH5^{macKO-Td}$ and $Cx3cr1Cre^{Td}$ mice, and then infected with empty vector or IL11 over-expression lentivirus ($10^8$ Pfu). Recipient C57BL/6 mice were subjected to 10-Gy lethal-dose irradiation from cobalt 60 source followed by tail vein injection of $5 \times 10^6$ congenic donor BM with or without IL11 over-expression. The chimeras were recovered for 2 weeks to reconstitute bone marrow, and then implanted osmotic mini-pumps (Alzet) containing either saline or angiotensin II (Meilunbio, MB1677-2) under back for 2 weeks. The recipient mice received autoclaved water containing $32 \times 104$ U/L Gentamicin Sulfate (Biosharp, Cat#: BS143) and 250 U/L Erythromycin (Aladdin, Cat#: E105345) 1 week before and until euthanasia by carbon dioxide ($CO_2$) inhalation for analysis after the BMT.

## Histology

The mice euthanasia was conducted using carbon dioxide ($CO_2$) inhalation, and the hearts were rinsed in PBS and fixed in 4% paraformaldehyde (PFA) for overnight. The hearts were then dehydrated and paraffin embedded, and sectioned at 4-μm interval. The sections were stained with Masson's Trichrome (Servicebio; Cat# G1006) for detection of cardiac fibrosis and Alexa FluorTM 488 conjugated wheat germ agglutinin (WGA) (W11261, Invitrogen) for measurement of cardiomyocyte size in vivo after deparaffinization and rehydration. Slides were imaged using the Leica DM IL inverted contrasting microscope and percentage heart fibrosis and cardiomyocyte size were quantified with Image J software.

## Immunofluorescent staining

The fixed hearts were frozen embedding and sectioned at 8-μm interval. The sections were permeabilized and blocked in PBS-T (0.02% triton X-100) with 1% goat serum for 1 hr. Immunostaining was performed using the following antibodies diluted in PBS-T (0.02% triton X-100) at 4 °C overnight: rabbit anti-SMA (Cell Signaling Technology; Cat# 19245, 1:500), rabbit anti-CD11b (Cell Signaling Technology, Cat#49420, 1:200), rabbit anti-Ki67 (Abcam; Cat# ab15580, 1:200). Sections were then incubated with 488 nm-conjugated donkey anti-rabbit secondary antibody (Invitrogen; Cat# A-21206, 1:500), 594 nm-conjugated donkey anti-rabbit secondary antibody (Invitrogen, Cat#: A21207, 1:500) following washing with PBS. Slices were mounted with vectashield mounting medium following DAPI staining and images were captured by fluorescent microscope. A section with the corresponding IgG of the primary antibody was used to histology control for background.

## Fluorescence Activated Cell Sorting (FACS)

The hearts were isolated and washed with Hanks Balanced Salt Solution (HBSS) for 3 times and then digested to single cell suspensions with Enzyme cocktail containing 125 U/mL collagenase Type XI (Sigma-Aldrich, C7657), 60 U/mL hyaluronidase (Sigma-Aldrich, H3506), 60 U/mL DNaseI (Sigma-Aldrich, 10104159001), 450 U/mL collagenase Type I (Sigma-Aldrich, SCR103) and 20 mM HEPES at 37 °C for 45 min.

For SMA detection, single cell suspensions from Td+ mice were fixed in 2% paraformaldehyde/PBS (v/v) for 20 min at 4 °C, and then permeabilized in 0.1% (w/v) saponin in PBS at 4 °C for 10 min. The mouse anti-SMA (Proteintech; Cat# 67735, 1:100) antibody was added in the cells and incubated overnight at 4 °C. The next day, cells were incubated with 647 nm-conjugated anti-mouse secondary antibody (Invitrogen; Cat# A56576, 1:100) and fluorescent labeled antibodies for macrophages in a dark environment at 4 °C for 30 minutes. The cell suspensions were then stained with anti-live and dead (BioLegend; Cat# 77184, 1:200), anti-CD11b (BioLegend; Cat# 550993, 1:100), anti-Lyve1 (Invitrogen; Cat# 50-0443-82, 1:100).

For single-cell suspensions from GFP+ mice, anti-CD11b (BioLegend; Cat# 101259, 1:100), and anti-Ly6c (BD Pharmingen™; Cat# 560525, 1:100) were used to detect macrophage.

The above-stained cells were then washed three times with 1xPBS before flow cytometry analysis. Data were acquired on NovoCyte 3000

flow cytometer (Agilent Technologies) and analyzed with NovoExpress software version 1.5.0.

## Peritoneal macrophage isolation

Briefly, 1 mL of 3% brewer thioglycollate broth was injected into the peritoneal cavity. After three days of injection, the macrophages were collected by injecting and aspirating 10 mL PBS from the peritoneum. The cells were replaced with fresh RPMI 1640 medium supplemented with 10% fetal bovine serum and cultured for later experiments.

## Primary cardiac fibroblast isolation and culture

The hearts were dissected and cut into small pieces and digested with 1 mg/ml Collagenase II in DMEM medium at 37 °C for 1 hour. Cells were passed through a 70 μm cell strainer (BD Falcon) and centrifuged. The cell pellet was re-suspended and cultured in DMEM containing 10% FBS.

## Immunocytochemistry

Briefly, Ang II-treated macrophages or macrophage condition medium-treated fibroblasts were fixed with 4% paraformaldehyde and permeated with 0.5% Triton X-100 at room temperature, and then incubated with rabbit anti-Ki67 (Abcam; Cat# ab15580, 1:500) or rabbit anti-SMA (Cell Signaling; Cat# 19245, 1:500) for overnight at 4 °C. The samples were mounted on glass slides after incubation with Alexa Fluor™594 donkey anti-rabbit IgG(H + L) (Invitrogen, Cat# A21207, 1:1000) or Alexa Fluor™ 488 Goat anti-Rabbit IgG (H + L) (Invitrogen, Cat#:A1100, 1:1000) secondary antibody, and the fluorescent images were obtained under ZEISS LSM 710 confocal microscope or Leica DM IL inverted contrasting microscope.

## Western blot

Cells or tissues were homogenized in cold RIPA buffer containing protease and phosphatase inhibitors. The lysates were collected following centrifugation and protein concentration was determined using a BCA protein concentration determination kit (Beyotime; Cat# P0011). The supernatant was stored at −80 °C transferred to a polyvinylidene difluuntil use. The proteins were separated by electrophoresis and oride membrane (Millipore, Billerica, MA, USA). The membranes were blocked with fat-free dry milk and then incubated with indicted primary antibodies as follows: rabbit anti-Col I (Abcam; Cat# ab138492, 1:1000), rabbit anti-Col III (Abcam; Cat# ab184993, 1:1000), rabbit anti-SMA (Cell Signaling; Cat# 19245, 1:2000), rabbit anti-ALKBH5 (Proteintech; Cat# 16837-1-AP, 1:1000), rabbit anti-IL11RA1 (Abcam; Cat# ab125015, 1:1000), rabbit anti-IL11 (Proteintech; Cat# 55169-1-AP, 1:1000) and rabbit anti-GAPDH (Proteintech; Cat# 10494-1-AP, 1:5000). The immuno-positive bands were visualized by enhanced chemiluminescence reagent (Millipore, USA) following incubation with a secondary peroxidase-conjugated anti-rabbit (Beyotime; Cat# A0208, 1:10000) or anti-mouse (Beyotime; Cat# A0216, 1:10000) antibody. Image J (Vl.8.0) analysis was used to quantify the bands of western blot images and GDPDH was used as an internal reference.

## RNA isolation and qPCR

Total RNA was isolated with an RNA Purification Kit at room temperature according to the manufacturer's instructions. For analysis of mRNA expression, 1 ug RNA was reverse-transcribed into cDNA with the cDNA synthesis kit (EZBioscience, United States). Quantitative real-time PCR analysis was then executed with SYBR green qPCR master mix on the Biorad PCR system (Thermo Fisher Scientific). GAPDH was taken as endogenous control. Each sample was run at least in triplicate. The primers used for qPCR analysis are listed in Supplementary Table 1.

## Lentiviral package

Mouse full-length open reading frames of IL11 was subcloned into pLenti-CMV-MCS-HA-3Flag vector. Human embryonic kidney (HEK) 293 T cells (ATCC, CRL-1573) were transfected with vector plasmid and the lentiviral packaging vectors pCMV.DR8 and pMD2.G (Addgene, Plasmid #12259). The viral supernatants were harvested on days 2 and 3 after transfection, filtered with 0.4-μm filters.

## RNA immunoprecipitation-sequencing

For immunoprecipitation-sequencing, total RNA was isolated from peritoneal macrophage. Briefly, macrophages with and without Ang II treatment were re-suspended and lysed in RIP lysis buffer, and the lysates were used as input. Then, lysates were incubated with ALKBH5 antibody in RIP immunoprecipitation buffer at 4 °C for 4 hours. Normal Rabbit IgG (Proteintech) was used as a negative control. The lysates were immunoprecipitated by Protein A/G beads (Repligen) at 4 °C for an additional 2 hours. The magnetic bead-bound complexes were centrifuged and washed for 5 times. The precipitated RNA samples were extracted and purified with phenol, chloroform and isoamyl alcohol, and the immunoprecipitated RNA was used to construct RNA libraries. Then, library sequencing was performed by illumine HiSeq 4000 sequencer with 150 bp paired-end reads. Data analysis was performed according to the published procedure.

## Methylated RNA immunoprecipitation sequencing and MeRIP-qPCR

Peritoneal macrophages were isolated from ALKBH5 knockout or control littermate mice, and total RNA was isolated with RNA Purification Kit (EZBioscience, B0004D, China) as above described. RNA samples were chemically fragmented into,100-nucleotide-long fragments by 5 min incubation at 94 °C in fragmentation buffer (10 mM ZnCl2, 10 mM Tris-HCl, pH 7). The fragmentation reaction was stopped with 0.05 M EDTA, followed by standard ethanol precipitation. Then the fragmented RNA (400 mg mRNA or 2.5 mg total RNA) was incubated for 2 hours at 4 °C with 5 mg of anti-m6A antibody (Abcam; Cat# ab151230) in IPP buffer (150 mM NaCl, 0.1% NP-40, 10 mM TrisHCl, pH 7.4) supplemented with RNase inhibitors. The mixture was then immunoprecipitated by Protein A/G beads (Repligen) at 4 °C for an additional 2 hours. After extensive washing, bound RNA was eluted from the beads with 0.5 mg/ml N6-methyladenosine (Sigma-Aldrich) in IPP buffer, and ethanol precipitated. RNA was resuspended in $H_2O$ and used for library generation with mRNA sequencing kit (Illumina). m6A enrichment was determined by qPCR analysis with primers in the Supplemental Table 1.

## m6A dot blot assay

Total RNA was isolated from the sorted Td+ cardiac macrophages with RNA Purification Kit (EZBioscience, B0004D, China). The serially diluted mRNA was denatured at 95 °C to disrupt secondary structures for 3 min, and then chilled on ice immediately. mRNA was dropped and crosslinked onto the Hybond-N+ membrane in Stratalinker 2400 UV Crosslinker under Autocrosslink mode (1200 microjoules [x100]; 25–50 sec). The membrane was washed in 10 ml of wash buffer in a clean washing tray for 5 min at room temperature. The membrane was blocked for 1 hour at room temperature, and then incubated with anti-m6A antibody (1:250 dilution; 2 μg/ml) overnight at 4 °C with gentle shaking. The membrane was incubated with IgG-HRP (1:10,000 dilution; 20 ng/ml) for 1 hour at room temperature with gentle shaking. The immuno-positive images were visualized by enhanced chemiluminescence reagent (Millipore, USA) according to the manufacturer's instructions. The membranes were then stained with 0.02% methylene blue in 0.3 M sodium acetate (pH 5.2) for 2 hours and washed with ribonuclease-free water for 5 hours.

## RNA stability assays

Macrophage were treated with actinomycin D at a final concentration of 5 μg/mL. Total RNA was extracted at 0, 1 and 2 hours after adding actinomycin D. The mRNA stability was determined by RT-qPCR and the data were normalized to that at $t = 0$ time point.

## Parabiosis study

*Cx3cr1Cre; Rosa26^Td* or *Cx3cr1Cre; Alkbh5^flox/flox; Rosa26^Td* donor mice were surgically joined with *CCR2^GFP/GFP* recipient mice with longitudinal incisions along the side of the mouse extending from approximately 1 cm behind the ear to just past the hind limb. Mice remained surgically joined for 7 days to connect the blood circulation before Angiotensin II infusion. Hearts were harvested at 14 days following Angiotensin II infusion and analyzed by Immunohistochemistry or flow cytometry.

## Non-myocardial single-cell RNA sequencing

**Single-cell isolation.** In brief, hearts from Angiotensin II treated C57BL/6 mice were isolated and stored in the sCelLiveTM Tissue Preservation Solution (Singleron Bio Com, Nanjing, China) on ice within 30 mins. The hearts were washed with Hanks Balanced Salt Solution (HBSS) 3 times and then digested with 2 ml sCelLiveTM Tissue Dissociation Solution (Singleron) by Singleron PythoN™ Automated Tissue Dissociation System (Singleron) at 37 °C for 15 mins. The cell solution was then passed through a 40 μm cell strainer to collect the non-CM cells, centrifuged at $500 \times g$ for 5 mins and suspended softly with GEXSCOPE® red blood cell lysis buffer (Singleron, 2 ml) for another 10 mins to remove red blood cells. Finally, the samples were stained with trypan blue (Sigma, United States) and the cellular viability was evaluated microscopically. The live single suspension was run on a BD LSRII machine and then constructed library as follows.

**scRNA sequencing.** Sorted single-cell suspensions ($1 \times 10^5$ cells/ml) with PBS (HyClone) were loaded into microfluidic devices using the Singleron Matrix® Single Cell Processing System (Singleron). Subsequently, the scRNA-seq libraries were constructed according to the protocol of the GEXSCOPE® Single Cell RNA Library Kits (Singleron)[48]. Individual libraries were diluted to 4 nM and pooled for sequencing. At last, pools were sequenced on Illumina HiSeq X with 150 bp paired end reads.

**Pre-processing and quality control for scRNA-seq data.** Raw reads were processed to generate gene expression profiles using an internal pipeline in Singleron company (Nanjing, China). Briefly, cell barcode and UMI was extracted after filtering read one without poly T tails. Adapters and poly-A tails were trimmed (fastp V1) before aligning read two to GRCh38 with ensemble version 92 gene annotation (fastp 2.5.3a and featureCounts 1.6.2)[10]. Reads with the same cell barcode, UMI and gene were grouped together to calculate the number of UMIs of genes in each cell. The UMI count tables of each cellular and barcode were employed for further analysis. Cell type identification and clustering analysis were performed by the Seurat program (Version 4.3.0)[49,50]. UMI count tables were loaded into R using read.table function. Afterwards, parameter resolution to 0.6 was set for FindClusters function to clustering analyses. Differentially expressed genes (DEGs) between different samples or consecutive clusters were identified with function FindMarkers[51]. RNA velocity analysis by scVelo was performed to analyze the origin of myofibroblasts.

**Statistical analysis.** All data were presented as mean ± SEM. Shapiro-Wilk test was used to determine the normality of data, and all data passed normality and equal variance. Comparisons between 2 groups were analyzed by 2-tailed Student's t-test. Differences between multiple groups were performed using one-way ANOVA followed by Tukey post-hoc tests. 2-way ANOVA followed by Tukey post-hoc tests was used for comparisons between multiple groups when there were 2 experimental factors. Biological experimental replicates in each group are shown in Fig. legends. $P < 0.05$ was considered to denote statistical significance. All statistical analyses were performed by GraphPad.Prism.9 software package (SPSS Inc, Chicago, IL).

## Reporting summary

Further information on research design is available in the Nature Portfolio Reporting Summary linked to this article.

## Data availability

All data are available in the main text or the supplementary materials. Source data are provided within this paper. The single cell RNA-sequencing and Methylated RNA immunoprecipitation sequencing data generated in this study have been deposited in the Genome Sequence Archive database under accession code CRA012231. Additional data related to this paper are available from the corresponding author on request. Source data are provided with this paper.

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

## Acknowledgements

This work was supported by the National Key R&D Program of China (2022YFA1104200 to C.-C. R), National Natural Science Foundation of China (82070456 to T.Z, 81922004 to C.-C. R, 32271233 to C.-C. R, 82300513 to Y.Lei., 92368105 to C.-C. R), Shanghai Rising-Star Program (21QA1401400 to T.Z.), Young Health Talents of Shanghai Municipal Health Commission (2022YQ069 to T.Z.), Shanghai Science and Technology Commission (21XD1420500 to C.-C. R), and the Innovative research team of high-level local universities in Shanghai and a key laboratory program of the Education Commission of Shanghai Municipality (ZDSYS14005 to C.-C. R).

## Author contributions

T.Z. designed and performed the experiments, analyzed the data, and wrote the manuscript. M.-H.C. designed and performed the experiments, analyzed the data. R.-X.W., J.W., X.-D. H., T.M. A.-H.W., Y.Lei., Y.-F.Y., Y. Li., D.-H. H. and Y.-X.L. performed the experiments. L.Z. and A.-J. S. provided suggestions on the project design. G.-N.Z., W.L. and J.-L.Z supervised scRNA sequencing data, provided the lipid nanoparticles, commented on the results and revised the manuscript. C.-C.R. conceived the project, designed and supervised the experiments, and wrote and revised the manuscript.

## Competing interests

The authors declare no competing interests.

## Ethics

The animal experiments were approved by the Ethics Committee of Fudan University and were performed in accordance with the National Institutes of Health (NIH) Guide for the Care and Use of Laboratory Animals.

## Additional information

[1]Department of Physiology and Pathophysiology, Shanghai Key Laboratory of Bioactive Small Molecules, State Key Laboratory of Medical Neurobiology, School of Basic Medical Sciences, and Jinshan Hospital, Fudan University, Shanghai, China. [2]Institute of Metabolism and Regenerative Medicine, Shanghai Sixth People's Hospital Affiliated to Shanghai Jiao Tong University School of Medicine, Shanghai, China. [3]Minhang Hospital and School of Pharmacy, Key Laboratory of Smart Drug Delivery Ministry of Education, State Key Laboratory of Molecular Engineering of Polymers, Fudan University, Shanghai, China. [4]Department of Cardiology, RuiJin Hospital/LuWan Branch, Shanghai Jiao Tong University School of Medicine, Shanghai, China. [5]Department of Critical Care Medicine, The First Affiliated Hospital of Nanjing Medical University, Nanjing, China. [6]Department of Cardiology and Institute for Developmental and Regenerative Cardiovascular Medicine, Xinhua Hospital, Shanghai Jiaotong University School of Medicine, Shanghai, China. [7]Department of Cardiology, Zhongshan Hospital, Fudan University, Shanghai Institute of Cardiovascular Diseases, Shanghai, China. [8]Department of Immunology, Nanjing Medical University, Nanjing, Jiangsu, China. [9]Department of Geriatrics, Ruijin Hospital, Shanghai Jiao Tong University School of Medicine, Shanghai, China. [10]These authors contributed equally: Tao Zhuang, Mei-Hua Chen. ✉e-mail: wlu@fudan.edu.cn; gnzhang@njmu.edu.cn; zuo-junli@163.com; ruancc@fudan.edu.cn

