## [Peer Review File · Nature Communications]

ALKBH5-mediated m6A modification of IL-11 drives macrophage-to-myofibroblast transition and pathological cardiac fibrosis in miceREVIEWER COMMENTS

Reviewer #1 (m6A, immune signaling) (Remarks to the Author):

This is an interesting study that uses mouse model of hypertension that leads to fibrosis. they observe ALKBH5 overexpression when leads to increased stability for IL11. knockout of alkbh5 in mice rescued the phenotype , while IL11 over-expression restored the disease in ALKBH5 knockout mice.

the study is well done, uses exacting knockout mice and over-expression alleles, and disease model is suitable and results

are convincing. the pinning down of IL11 involvement is impressive the rescue experiment by IL11 overexpression boosts the conclusions made.

The paper is suitable for publication and i found only minor comments for clarity:

1) sometimes the language usage is confusing - for example when authors say "alkbh5 mediate m6A effect on IL-11".. they should be clear ..what was the mediated effect reduced signal or increased signal..for the non expert reader this can be confusing and hard to comprehend and the authors should always describe that effect positive or negative, decrease or increase...so that the reader can easily follow

2) the authors show in IGV panel only the m6A profile on IL11 gene in WT vs. KO..but i think profile of other positive and negative genes should be shown as the signal change is weak for IL-11!. how sure the authors that the phenotype is predominantly mediated by IL11 pathway and not other candidate pathways that are synergistic?

Reviewer #2 (Cardiac fibrosis/inflammation, Ang-II model) (Remarks to the Author):

Cardiac macrophage biology is emerging as an important facet within the context of the pathogenesis of cardiac fibrosis. That said, the spectrum of macrophage phenotypes and associated function tied to each phenotype is not well studied. The authors have carried out single-cell transcriptomics, in vivo and in vitro cell tracing and parabiosis to interrogate

whether circulating monocyte derived cardiac macrophages in some way influence myofibroblast activation in the context of Ang II-induced hypertension. The authors were interested in RNA N6-methyladenosine demethylase ALKBH5 expression, as they found that it was increased in macrophage-to-myofibroblast transition. Knocking out this gene was associated with improved cardiac fibrosis and cardiac dysfunction in Ang II treated animals, and they elucidated the mechanism of action of this gene. Further the intervention of ALKBH5 or IL-11 receptor α (IL11RA1) siRNA conferred an anti-fibrotic effect. The authors have interpreted their data appropriately and the writing is well done. With the following comments taken into consideration the impact of the paper will be increased.

Specific comments.

1. The abstract, introduction, methods and results sections are well written, but will require some editing to fine tune word usage in plurality and tenses. Repeated terms such as “cell fate decision” implies cellular sentience in the abstract and discussion, and this may be improved by restating as “factors that regulate the phenotype or transition and activation of cardiac macrophages” or something that aligns with the direction of the paper is recommended. “We herein provide” might be best restated as “Herein, we provide”, etc... some extra work is needed to smooth out the prose and readability.
2. The collected experiments are nicely conceived and executed. The paper highlights a growing awareness of the importance of the proinflammatory state in the pathogenesis of heart failure, and in particular the role of cardiac fibrosis and inflammation. The excitement of the main finding eg, in determining a specific role for RNA N6-methyladenosine demethylase ALKBH5 expression in MMT is obvious to the reader. The finding is novel. However, a number of papers have documented the importance of resident fibroblasts as a major pool for activated myofibroblasts in various etiologies of cardiac diseases with fibrosis. Did the authors consider the potential role of RNA N6-methyladenosine demethylase ALKBH5 expression, and IL-11 etc in cardiac resident quiescent fibroblasts (non activated and hyposynthetic cells) to activated hypersynthetic myofibroblasts? Is it possible that simple fibroblast activation is working in combination with MMT in their model of fibrosis?
3. aSMA (acta2) positive cells are usually myofibroblasts, but may only be truly labelled as

such with the concomitant formation of stress fibres eg, aSMA positive cells may be inactive fibroblasts if they don't feature stress fibres as well. Fig 2G and Fig 2H (the Western) provide some evidence that both the increased incidence of stress fibres and elevated aSMA is present in higher abundance in the WT cells vs the ALKBH5macKO cells, but the images in Fig 2G might be improved or enlarged somewhat to really underscore this difference.

_ - Ian Dixon

Reviewer #3 (Macrophage, cardiac inflammation) (Remarks to the Author):

In this report, the authors address the contribution of macrophages to pro-fibrotic myofibroblasts following ANGII cardiac injury. Using a variety of in vivo approaches, importantly including fate-mapping and parabiosis strategies, the authors suggest that cardiac macrophages (particularly monocyte derived) were able to contribute to the myofibroblast pool. Mechanistically the authors link de-differentiation with modifications of gene regulation of IL-11, performed by ALKBH5. While many of the approaches are interesting, numerous concerns exist regarding the reporting and interpretation of results. Many requests are below to allow for an improved understanding of the data.

The inclusion of additional approaches, such as bone marrow chimera experiments, adoptive transfer, or arranging new models, to definitely test the hypothesis and/or alternative hypotheses would be helpful. Specifically, if the authors intend to maintain that macrophages contribute to myofibroblast differentiation, they should also test the opposing hypothesis that non-macrophages are differentiating into this population. Specifically, in many other systems, SM cells are known to upregulate macrophage-associated gene programs during inflammatory responses.

Major Comments:

- Figure 1C: the authors do not define the shared features of the macrophage cluster. It would be helpful to show that these cells express canonical macrophage genes.
- Fig s1: Acta+ scRNA-seq cells don't appear to express Adgre1. CD68 is not a great discriminating marker for macrophage lineage. Do these cells express other known cardiac

macrophage genes *Fcgr1* or *Csf1r*?

-Fig 1F/ s3: Can the authors please show tdTomato labeling and in depth gating information for flow cytometry data. The current data is not sufficient to determine labeling efficiencies within cell subsets.

-Fig 2A: heat maps should be presented as log-fold change. Not mean expression of the group, which can be misleading. Also provide statistics with Adjusted P-value or FDR for this analysis.

-Line 126: The Authors conclude that since a subset of SMA-like cells expressed tdTomato at analysis that it was because macrophages transitioned to SMA cells. However, this experiment is supportive of multiple interpretations. It also allows for the potential that SMA-expressing cells turn on macrophage-like genes to express tdTomato. Since the fate-mapping strategy using Cre-ERT2 failed to show contributions to the SMA-expressing cells, it seems that the second hypothesis would be more suited by the data. If the authors truly wanted to answer this question, they would perform SM-cell fate mapping using the Myh11-creERT2 mouse model and also attempt to perform adoptive transfer or parabiosis studies.

-The data shown in Figure 2B does not support the author's conclusion that *Alkbh5* is upregulated during pseudotime. Quantification and statistics should be shown to help draw conclusions. *Alkbh5* should also be shown in expression dot plots of scRNA-seq data in the total cluster plot, as well as the macrophage subclustering.

-Figure 2C needs a loading control and this assay was also poorly described. How were macrophages specifically enriched and purity confirmed for dot-plot analysis? Since the cell subclusters change so dramatically, this type of bulk-assay is difficult to interpret.

-Fig s4A: how do the authors interpret the data that the percentage of *Alkbh5* expressing macrophages reduces across the pseudotime toward myofibroblast? In a prior section the authors claimed that macrophages upregulated *Alkbh5* across pseudotime. Oddly, data from qRT-PCR suggested expanded ALKBH5 expression in bulk analysis in s4B. This data should also report CT information for each analysis, and the CT data for normalization.

-Figure 2D: please show all blots used for quantification of this data. The current blots do not appear to have 2x protein shifts.

-Figure 2E: Reporting only percentage data is very difficult to interpret. It appears that there is a dramatic overall reduction in tdTomato+ cells. Does *Alkbh5* deletion in macrophages

result in changes in cardiac macrophage numbers in the steady state or following challenge? Which subsets of macrophages are most influenced by Alkbh5-deletion? What about blood monocytes?

-It would be helpful to understand the macrophage response following IL-11-OE. Data for macrophage numbers and phenotype by flow cytometry should be shown for the full experiments in control and challenged mice (WT, IL1OE, KO, KO IL11OE), such as in Figure 6E/F.

-Figure 4: It is surprising that a short-term parabiosis study would incite such dramatic changes in the heart injury model, given the expected minor contribution of chimeric cells to the cardiac myeloid pool. Previously published studies suggested that ~30% of circulating monocytes will come from the parabiont and since the majority of tissue resident macrophages are long-lived/self-maintained, it's surprising that much more than a few percentage of cardiac macrophages would be from the donor. Could the authors discuss how such a minor proportion of cells might inhibit cardiac inflammation? It seems like this experiment would be dramatically more likely to work in a CCR2^{-/-} pair, where all circulating monocytes could come from the donor.

-Figure 4: necessary control data is missing that would allow for interpretation of this experiment. A) blood chimerism needs to be reported after chimerism, b) cardiac macrophage replacement from donor cells should be reported in untreated and ANG-II treated mice, and c) flow cytometry needs to be performed to assess changes in macrophage phenotype comparing tdTomato⁺ and tdTomato⁻ subsets within the heart. Again, imaging shows reduced tdTomato⁺ cells, suggesting that intrinsic changes in macrophage numbers may be playing a role in the activation of SM cells.

-Deletion of ALKBH5 leads to decreased macrophage proliferation and reduced activation of SM and Fibroblasts in co-culture assays. These data clearly show that cardiac macrophages rely on ALKBH5 for normal inflammatory function. Furthermore, it makes analysis of the proposed MMT profile of these cells difficult to assess using this model.

-Figure 5A: Please report adjusted P-value and FDR for macrophage RIP-seq study. Also, unbiased analysis and full datasets from the gene expression analysis need to be shared. Top 10 pathways enriched would also help to understand the macrophage response to ANG-II.

- Figure S7: Please also show control and knockout data for mice that were not treated with

ANG-II. Gating approach is disturbing and brings into question the gating used across the manuscript. Are the authors solely generating flow cytometry gates based on tdTomato-negative samples? While this would be potentially acceptable for antibody staining, it is a mistake for reporter mice, particularly when using inflammatory models. This is because cells within tissue often bleb or apoptosis, and small cellular components can be taken up by neighboring cells (this is not restricted to primary phagocyte lineages). Thus, a modest level of tdTomato can often be detected in cells that are indeed tdTomato-negative. There is an obvious cut-off between the 3-4 log of the data where it seems to be much more appropriate to perform this analysis. Did CX3CR1-creERT2 ALKBH5-flox mice show similar tdTomato+ and tdTomato- macrophage numbers compared with controls? It would be expected that a proliferation defect in the ALKBH5-deleted cells may lead to dramatic replacement of these cells during the “tamoxifen-rest” period, which would be evident by a reduced number of Tomato-cells in the ALKBH5-deleted hearts, even in the absence of injury.

-LNP assays should show biodistribution and whether the therapeutic has effects in other tissues and cell types.

Minor Comments:

-overall the writing needs to be toned down. The authors often write “confirm, show, validate, etc” language, when it is more appropriate to say “suggests, supports, or infers”. An example is when discussing pseudotime trajectory analysis between clusters (Line 55), the data “suggests” there may be a link between macrophage and a fibroblast subset. It does not show that one exists, and the authors should emphasize this point to justify why it needs to be tested rigorously in multiple experimental models. In addition, many of the data shown have multiple interpretations and should be included in the paper – this is particularly true with MMT data. The majority of MMT data would also support that a subset of SM cells are upregulating macrophage-associated markers.

-Sex as a biological variable is not discussed appropriately.

-Figures need to be reported in order they are presented in the text. An example is that Fig 2D is discussed before 2A-C.

-Authors claim to have sorted non-cardiomyocytes for scRNA-seq. However, in the methods they simply filtered cells through a 40um strainer. Please describe what was actually

performed in the text. “Cells were filtered through a 40um filter to enrich for non-CM cells”. To state the cells were sorted mislead the reader and also confuses us considering that CMs are present in the scRNA-seq analysis (Fig 1A).

REVIEWER COMMENTS

Reviewer #1 (m6A, immune signaling) (Remarks to the Author):

This is an interesting study that uses mouse model of hypertension that leads to fibrosis. they observe ALKBH5 overexpression when leads to increased stability for IL11. knockout of alkbh5 in mice rescued the phenotype, while IL11 over-expression restored the disease in ALKBH5 knockout mice.

the study is well done, uses exacting knockout mice and over-expression alleles, and disease model is suitable and results are convincing. the pinning down of IL11 involvement is impressive the rescue experiment by IL11 overexpression boosts the conclusions made.

Response:

Thank you for your kind comments that our study is interesting, well done and suitable for publication.

The paper is suitable for publication and i found only minor comments for clarity:

1) sometimes the language usage is confusing - for example when authors say "alkbh5 mediate m6A effect on IL-11".. they should be clear ..what was the mediated effect reduced signal or increased signal..for the non expert reader this can be confusing and hard to comprehend and the authors should always describe that effect positive or negative, decrease or increase...so that the reader can easily follow

Response:

Thank you for your suggestion, we clarified the description as “ALKBH5-decreased m6A modification on IL-11 mRNA increases the protein level of IL-11, that drives macrophage-to-myofibroblast transition and pathological cardiac fibrosis” in the Abstract section and in line 12 at page 3 in the Result section.

2) the authors show in IGV panel only the m6A profile on IL11 gene in WT vs. KO..but i think profile of other positive and negative genes should be shown as the signal change is weak for IL-11!. how sure the authors that the phenotype is predominantly mediated by IL11 pathway and not other candidate pathways that are synergistic?

Response:

Thank you for your supportive suggestions. In fact, Our ALKBH5 RNA immunoprecipitation-sequencing (RIP-seq) data revealed that several genes are the direct targets of ALKBH5. For example, MMP9, which has been reported to be regulated by ALKBH5 in cancers (Cancer Biol Ther. 2023 Dec 31;24(1):2249174.), might also be the direct target gene of ALKBH5 in macrophages (Table S2). We provided the IGV panel showing m6A on MMP9 gene (Response Figure 1A) as the positive gene. We also provided the negative gene Arg1, that was exist in RIP-seq data but not in m6A sequencing profile (Response Figure 1B). Although the signal change for IL-11 is weak in m6A sequencing profile, the fold change of IL-11 in ALKBH5 RIP-seq is at the top of the list (Figure 5A). Therefore, we undertook comprehensive

considerations of m6A sequencing and RIP-seq, and found that IL11 was the critical candidate gene for the phenotype.

To further confirm the role of IL11, we inhibited IL11 signaling in cultured macrophages using neutralizing antibody against IL11 in the revised manuscript. As shown in the revised Figure S9G-S9I, inhibition of IL11 signaling reduced SMA and Col1 expression and cell proliferation in WT macrophages, but had no effect on ALKBH5 knockout macrophages. These data suggest that IL-11 is a critical target for ALKBH5 in this model. However, we could not exclude the other ALKBH5 downstream candidate genes that may be involved in the regulation of this phenotype in the macrophages. This needs further more investigation in the future. The revised text is in lines 6-9 at page 13, lines 20-27 at page 16, and the new data were added in the Figure S9G-S9I.

Response Figure 1

Response Figure 1: m6A peak distribution on MMP9 and Arg1. **A**, m6A peak distribution on MMP9 mRNA shown by Integrative Genomics Viewer (IGV) tool. kb means kilo base pair. **B**, m6A peak distribution on Arg1 mRNA shown by Integrative Genomics Viewer (IGV) tool. kb means kilo base pair.

Reviewer #2 (Cardiac fibrosis/inflammation, Ang-II model) (Remarks to the Author):

Cardiac macrophage biology is emerging as an important facet within the context of the pathogenesis of cardiac fibrosis. That said, the spectrum of macrophage phenotypes and associated function tied to each phenotype is not well studied. The authors have carried out single-cell transcriptomics, in vivo and in vitro cell tracing and parabiosis to interrogate whether circulating monocyte derived cardiac macrophages in some way influence myofibroblast activation in the context of Ang II-induced hypertension. The authors were interested in RNA N6-methyladenosine demethylase ALKBH5 expression, as they found that it was increased in macrophage-to-myofibroblast transition. Knocking out this gene was associated with improved cardiac fibrosis and cardiac dysfunction in Ang II treated animals, and they elucidated the mechanism of action of this gene. Further the intervention of ALKBH5 or IL-11 receptor α (IL11RA1) siRNA conferred an anti-fibrotic effect. The authors have interpreted their data appropriately and the writing is well done. With the following comments taken into consideration the impact of the paper will be increased.

Response:

We are appreciated for your comment that we have interpreted their data appropriately and the writing is well done.

Specific comments.

1. The abstract, introduction, methods and results sections are well written, but will require some editing to fine tune word usage in plurality and tenses. Repeated terms such as “cell fate decision” implies cellular sentience in the abstract and discussion, and this may be improved by restating as “factors that regulate the phenotype or transition and activation of cardiac macrophages” or something that aligns with the direction of the paper is recommended. “We herein provide” might be best restated as “Herein, we provide”, etc... some extra work is needed to smooth out the prose and readability.

Response:

Thank you for your great suggestions. We changed the “cell fate decision of cardiac macrophages” as “factors that regulate the phenotype or transition and activation of cardiac macrophages” in the Abstract, line 49 at page 2 in the introduction, and lines 4-5 at page 14 in the discussion sections. “We herein provide” were also restated as “Herein, we provide”.

2. The collected experiments are nicely conceived and executed. The paper highlights a growing awareness of the importance of the proinflammatory state in the pathogenesis of heart failure, and in particular the role of cardiac fibrosis and inflammation. The excitement of the main finding eg, in determining a specific role for RNA N6-methyladenosine demethylase ALKBH5 expression in MMT is obvious to the reader. The finding is novel. However, a number of papers have documented the importance of resident fibroblasts as a major pool for activated myofibroblasts in various etiologies of cardiac diseases with fibrosis. Did the authors consider the potential role of RNA N6-methyladenosine demethylase ALKBH5 expression, and IL-11 etc in cardiac resident quiescent fibroblasts (non activated and hyposynthetic cells) to activated hypersynthetic myofibroblasts? Is it possible that simple fibroblast activation is working in combination with MMT in their model of fibrosis?

Response:

Thank you for your comments that the collected experiments are nicely conceived and executed and our finding is novel. We very agree with the reviewer that fibroblast activation is working in combination with MMT in the cardiac fibrosis process. In fact, we performed cell co-culture assay for macrophages and cardiac fibroblasts by utilizing Transwell chamber (the revised Figure S9), the results indicated that the ALKBH5 KO in macrophages attenuated AngII-induced cardiac myofibroblast activation and proliferation. While IL-11 recombinant protein (rIL-11) rescued these effects of macrophage-mediated paracrine role on fibroblasts. As your supportive comments, we assessed the expression of ALKBH5, IL-11 and IL11RA1 in the cultured fibroblasts. The result showed that Ang II treatment had no significant effect on ALKBH5 and IL-11 expression, but increased IL11RA1 expression in cardiac fibroblasts (Response Figure 2). Taken together with the co-cultured assay, we suppose that ALKBH5/IL-11 pathway activation is mainly existed in cardiac macrophages. Then the secreted IL-11 induces MMT, as well as promotes cardiac fibroblast activation via directly binding to the receptor IL11RA1 in both macrophages and fibroblasts, since we detected IL11RA1

upregulation in both cells after Ang II treatment (Response Figure 2 and Figure 5J).

Response Figure 2

Response Figure 2: ALKBH5 in fibroblasts is not changed following Ang II treatment. **A**, mRNA levels of ALKBH5, IL11 and IL11RA1 expression in cultured fibroblasts with and without Ang II treatment. Error bars indicate mean \pm SEM. n = 5. n.s. indicates nonsignificant. **P < 0.01. **B**, Expression of ALKBH5, IL11 and IL11RA1 by western blot in cultured fibroblasts with and without Ang II infusion, with quantification at right. Error bars indicate mean \pm SEM. n = 5. **P < 0.01.

3. aSMA (acta2) positive cells are usually myofibroblasts, but may only be truly labelled as such with the concomitant formation of stress fibres eg, aSMA positive cells may be inactive fibroblasts if they don't feature stress fibres as well. Fig 2G and Fig 2H (the Western) provide some evidence that both the increased incidence of stress fibres and elevated aSMA is present in higher abundance in the WT cells vs the ALKBH5macKO cells, but the images in Fig 2G might be improved or enlarged somewhat to really underscore this difference.

Response:

Thank you for your great comments. We provided new immunofluorescent images of SMA+ staining in cultured Td+ macrophages with control PBS treatment. The data in Fig 2G and Fig 2H showed that Ang II increased SMA+ stress fibres in WT macrophages, and ALKBH5 knockout reversed Ang II-induced stress fibres and aSMA. The new data were added in Figure 2I.

_ - Ian Dixon

Reviewer #3 (Macrophage, cardiac inflammation) (Remarks to the Author):

In this report, the authors address the contribution of macrophages to pro-fibrotic myofibroblasts following ANGIO cardiac injury. Using a variety of in vivo approaches, importantly including fate-mapping and parabiosis strategies, the authors suggest that cardiac macrophages (particularly monocyte derived) were able to contribute to the myofibroblast pool. Mechanistically the authors link de-differentiation with modifications of gene regulation of IL-11, performed by ALKBH5. While many of the approaches are interesting, numerous concerns exist regarding the reporting and interpretation of results. Many requests are below to allow for an improved understanding of the data.

The inclusion of additional approaches, such as bone marrow chimera experiments, adoptive transfer, or arranging new models, to definitely test the hypothesis and/or alternative hypotheses would be helpful. Specifically, if the authors intend to maintain that macrophages contribute to myofibroblast differentiation, they should also test the opposing hypothesis that non-macrophages are differentiating into this population. Specifically, in many other systems, SM cells are known to upregulate macrophage-associated gene programs during inflammatory responses.

Response:

Thank you for your comment that our approaches are interesting and supportive suggestions. In the present study, we found that circulating monocyte-derived macrophages had a tendency of myofibroblast transition by utilizing lineage tracing (Figure 1F and 1G), bone marrow transplantation (Figure 6), parabiosis (Figure 4). As your comments, to determine the role of SM in this pathological process, we further utilized *Myh11Cre^{ERT2}; Rosa26^{Td}* mice, which trace SMA+ SM cells in the steady heart, to test whether SM cells upregulated macrophage-associated gene programs. FACS and immunostaining of CD11b and SMA in hearts of *Myh11Cre^{ERT2}; Rosa26^{Td}* mice revealed Td+SMA+ cells, but not Td+CD11b cells were increased in hypertensive hearts. Although we observed an increased CD11b+ cells in hypertensive hearts, these cells were not derived from Td+ cells (Figure S3E-S3G). These demonstrated that Myh11+ SM cells in the heart do not contribute to macrophage-like cells in this pathological process. The new data were added in the Figure S3E-S3G. The revises text is lines 31-38 at page 4.

Figure S3

Figure S3: E, Representative images of flow cytometry analyses of CD11b+ and SMA+ cells gated on Td+ cells in hearts *Myh11Cre^{ERT2}; Rosa26^{Td}* mice with control PBS or Ang II treatment. F-G, Representative immunofluorescent images of SMA (F) and CD11b (G) in hearts from PBS and Ang II treated *Myh11Cre^{ERT2}; Rosa26^{Td}* mice

(n=4), with quantification at right. Error bars indicate mean \pm SEM. n.s. not significant. Scale bar, 100 μ m.

Major Comments:

- Figure 1C: the authors do not define the shared features of the macrophage cluster. It would be helpful to show that these cells express canonical macrophage genes.

-Fig s1: Acta+ scRNA-seq cells don't appear to express Adgre1. CD68 is not a great discriminating marker for macrophage lineage. Do these cells express other known cardiac macrophage genes Fcgr1 or Csf1r?

Response:

Since these two comments are closely related, we address them together. According to the reviewer's suggestion, we added feature plots of Fcgr1 or Csf1r in Figure S1B, and showed that Acta2+ cells express macrophage genes Fcgr1 or Csf1r. We agree with the reviewer that Acta2+ scRNA-seq cells don't express Adgre1, as well as reduced expression of the other macrophage marker genes. Therefore, we hypothesized that these macrophages may trans-differentiate into myofibroblasts.

-Fig 1F/ s3: Can the authors please show tdTomato labeling and in depth gating information for flow cytometry data. The current data is not sufficient to determine labeling efficiencies within cell subsets.

Response:

Thank you very much for your suggestions. We added Td+ labeling and in depth gating information for flow cytometry in Figure 1 and Figure S3. We also added gating information for flow cytometry analysis in the revised Figure S2A.

-Fig 2A: heat maps should be presented as log-fold change. Not mean expression of the group, which can be misleading. Also provide statistics with Adjusted P-value or FDR for this analysis.

Response:

Thank you for your supportive suggestions. We provided new heat maps (Figure 2B) presented as log-fold change, and also Violin image (Figure 2D) showing ALKBH5 expression in subclusters of macrophage with Adjusted P-value between macrophages and myofibroblasts in the revised Figure 2. The revised text is in lines 8-9 at page 5 and lines 1-4 at page 6.

-Line 126: The Authors conclude that since a subset of SMA-like cells expressed tdTomato at analysis that it was because macrophages transitioned to SMA cells. However, this experiment is supportive of multiple interpretations. It also allows for the potential that SMA-expressing cells turn on macrophage-like genes to express tdTomato. Since the fate-mapping strategy using Cre-ERT2 failed to show contributions to the SMA-expressing cells, it seems that the second hypothesis would be more suited by the data. If the authors truly wanted to answer this question, they would perform SM-cell fate mapping using the Myh11-creERT2 mouse model and also attempt to

perform adoptive transfer or parabiosis studies.

Response:

Thank you for the insightful comment and as the response above. We firstly performed new lineage tracing assay by utilizing *Myh11Cre^{ERT2}; Rosa26^{Td}* mice, which trace SMA+ SM cells in the steady heart, to test whether SM cells upregulated macrophage-associated gene programs. Immunostaining of CD11b and SMA in hearts of *Myh11Cre^{ERT2}; Rosa26^{Td}* mice revealed that Td+SMA+ cells, but not Td+CD11b cells were increased in hypertensive hearts (Figure S3E-S3G). Although we observed an increased CD11b+ cells in hypertensive hearts, these cells were not derived from Td+ cells.

We then performed parabiosis experiments by conjoining non-fluorescent C57BL/6/J mice with *Myh11Cre^{ERT2}; Rosa26^{Td}* mice, to test whether the donor Td+ SM Cells could move to the heart of recipient non-fluorescent C57BL/6/J mice, and detected macrophage-associated gene programs in Td+ cells during inflammatory responses. We observed few Td+ cells in hearts from both PBS and Ang II treated C57BL/6/J mice assessed by FACS analysis (Figure SH). These demonstrated that Myh11+ SM cells in the heart do not contribute to macrophage-like cells in this pathological process. The new data were added in the Figure S3E-S3H, and the revised text is lines 2-15 at page 4.

Figure S3

Figure S3: E, Representative images of flow cytometry analyses of CD11b+ and SMA+ cells gated on Td+ cells in hearts from *Myh11Cre^{ERT2}; Rosa26^{Td}* mice (n=5) with control PBS or Ang II. F-G, Representative immunofluorescent images of SMA (F) and CD11b (G) in hearts from PBS and Ang II treated *Myh11Cre^{ERT2}; Rosa26^{Td}* mice (n=4), with quantification at right. Error bars indicate mean \pm SEM. n.s. not significant. Scale bar, 100 μ m. H, Representative images of flow cytometry analyses of Td+ cells gated on live cells in hearts from C57BL/6 mice conjoined with *Myh11Cre^{ERT2}; Rosa26^{Td}* mice (n=5) with control PBS or Ang II.

-The data shown in Figure 2B does not support the author's conclusion that *Alkbh5* is upregulated during pseudotime. Quantification and statistics should be shown to help draw conclusions. *Alkbh5* should also be shown in expression dot plots of scRNA-seq data in the total cluster plot, as well as the macrophage subclustering.

Response:

Thank you for your critical comment. In fact, this is a visual error in Figure 2B, because ALKBH5 high expression cells (dark blue dots) was at upper layer, while ALKBH5 low expression cells (bright yellow dots) was at the lower layer. We have re-constructed this figure and showed that ALKBH5 was up-regulated in myofibroblast-like macrophages, which was in parallel with *Acta2* (the revised Figure 2C). The increased ALKBH5 expression in myofibroblast-like macrophages was also shown by Violin (the revised Figure 2D) and heat map (the revised Figure 2B). Besides, we provided ALKBH5 gene signatures in expression dot plots of scRNA-seq data in the total cluster plot in Figure S4A, as well as the macrophage subclustering in Figure 2A. All data showed that ALKBH5 was widely expressed in all cell clusters in the heart with more variation in macrophage cluster (Fig. S4A). Moreover, and a higher ALKBH5 expression was observed in myofibroblasts compared with macrophage populations. The revised text is in lines 6-9 at page 5 and lines 1-3 at page 6.

-Figure 2C needs a loading control and this assay was also poorly described. How were macrophages specifically enriched and purity confirmed for dot-plot analysis? Since the cell subclusters change so dramatically, this type of bulk-assay is difficult to interpret.

Response:

Thank you for the suggestion. We apologize for lacking loading control and poorly described results. We re-performed methylene blue staining and dot-plot in the revised Figure 2E. Actually, RNA m⁶A dot blot assays were performed in sorted Td⁺ cardiac macrophages from PBS or Ang II-treated *Cx3cr1Cre; Rosa26^{Td}* mice. The revised data were added in Figure 2E, and the revised text is lines 17-18 and lines 28-30 at page 21 at the Methods section.

-Fig s4A: how do the authors interpret the data that the percentage of *Alkbh5* expressing macrophages reduces across the pseudotime toward myofibroblast? In a prior section the authors claimed that macrophages upregulated *Alkbh5* across pseudotime. Oddly, data from qRT-PCR suggested expanded ALKBH5 expression in bulk analysis in s4B. This data should also report CT information for each analysis, and the CT data for normalization.

Response:

Thank you for your comments. Compared to the left 2 *Il6 α* ⁺ macrophage clusters, the fraction of cells and mean expression of ALKBH5 in the middle *Il6 α* ⁺*Ccn2*⁺ macro-myofibroblast cluster and the right 2 clusters were significantly increased. As we mentioned above (response to Fig 2A), ALKBH5 expression in macro-myofibroblast cluster is higher than macrophage clusters (the revised Figure 2B-2D). The revised text

is in lines 6-9 at page 5 and lines 1-4 at page 6.

As your kind comment, we provided CT information for each analysis below. Real-time qPCR was performed with housekeeping gene Gapdh as a control. Desired genes were normalized to endogenous housekeeping gene using $2^{-\Delta\Delta C_t}$ method.

CT	Mettl3	Mettl14	WTAP	FTO	ALKBH5	YTHDC1	YTHDC2	YTHDF1	YTHDF2	YTHDF3	HNRNPA2B1	Gapdh
PBS	27.05	27.89	29.98	31.81	30.64	28.71	27.81	17.48	17.31	17.96	26.11	22.41
	27.32	27.37	29.94	31.23	30.02	29.18	27.55	17.48	17.35	18.84	26.72	22.91
	27.28	27.26	29.74	31.76	30.56	29.10	26.41	18.08	18.16	17.49	26.92	22.14
Average	27.22	27.51	29.89	31.60	30.40	29.00	27.26	17.68	17.60	18.10	26.58	22.48
CT	Mettl3	Mettl14	WTAP	FTO	ALKBH5	YTHDC1	YTHDC2	YTHDF1	YTHDF2	YTHDF3	HNRNPA2B1	Actin
Ang II	26.50	26.06	29.41	30.98	27.11	27.88	26.53	18.07	17.83	18.94	26.41	22.17
	26.24	26.36	28.97	30.81	27.25	27.95	26.75	17.76	17.87	18.74	26.16	22.41
	26.28	26.89	29.26	30.57	27.56	26.66	25.96	17.59	17.71	18.71	25.73	21.70
Average	26.34	26.44	29.22	30.79	27.31	27.50	26.41	17.81	17.80	18.80	26.10	22.09
d-CT	Mettl3	Mettl14	WTAP	FTO	ALKBH5	YTHDC1	YTHDC2	YTHDF1	YTHDF2	YTHDF3	HNRNPA2B1	
PBS	4.56	5.42	7.50	9.33	8.15	6.23	5.33	-5.00	-5.17	-4.52	3.63	
	4.84	4.89	7.47	8.75	7.54	6.70	5.07	-5.00	-5.13	-3.64	4.24	
	4.80	4.79	7.27	9.28	8.08	6.62	3.93	-4.40	-4.32	-4.99	4.44	
Average	4.73	5.03	7.41	9.12	7.92	6.52	4.78	-4.80	-4.88	-4.38	4.10	
d-CT	Mettl3	Mettl14	WTAP	FTO	ALKBH5	YTHDC1	YTHDC2	YTHDF1	YTHDF2	YTHDF3	HNRNPA2B1	
Ang II	4.41	3.98	7.33	8.90	5.02	5.79	4.44	-4.02	-4.26	-3.15	4.32	
	4.14	4.28	6.89	8.72	5.16	5.86	4.66	-4.33	-4.22	-3.35	4.07	
	4.19	4.81	7.18	8.49	5.47	4.57	3.87	-4.50	-4.38	-3.38	3.64	
Average	4.25	4.35	7.13	8.70	5.22	5.41	4.32	-4.28	-4.29	-3.29	4.01	
dd-CT	Mettl3	Mettl14	WTAP	FTO	ALKBH5	YTHDC1	YTHDC2	YTHDF1	YTHDF2	YTHDF3	HNRNPA2B1	
PBS	-0.17	0.39	0.09	0.21	0.23	-0.29	0.55	-0.20	-0.29	-0.14	-0.47	
	0.11	-0.14	0.06	-0.37	-0.38	0.18	0.29	-0.20	-0.25	0.74	0.14	
	0.07	-0.24	-0.14	0.16	0.16	0.10	-0.85	0.40	0.56	-0.61	0.34	
Average	0.00	0.00	0.00	0.00	0.00	0.00	0.00	0.00	0.00	0.00	0.00	
dd-CT	Mettl3	Mettl14	WTAP	FTO	ALKBH5	YTHDC1	YTHDC2	YTHDF1	YTHDF2	YTHDF3	HNRNPA2B1	
Ang II	-0.32	-1.05	-0.08	-0.22	-2.90	-0.73	-0.34	0.78	0.62	1.23	0.22	
	-0.59	-0.75	-0.52	-0.40	-2.76	-0.66	-0.12	0.47	0.66	1.03	-0.03	
	-0.54	-0.22	-0.23	-0.63	-2.45	-1.95	-0.91	0.30	0.50	1.00	-0.46	
Average	-0.48	-0.68	-0.28	-0.42	-2.70	-1.11	-0.46	0.52	0.59	1.09	-0.09	
Fold change	Mettl3	Mettl14	WTAP	FTO	ALKBH5	YTHDC1	YTHDC2	YTHDF1	YTHDF2	YTHDF3	HNRNPA2B1	

PBS	1.12	0.77	0.94	0.86	0.85	1.22	0.68	1.15	1.22	1.10	1.38	
	0.93	1.10	0.96	1.29	1.30	0.88	0.82	1.15	1.19	0.60	0.91	
	0.95	1.18	1.10	0.89	0.90	0.93	1.81	0.76	0.68	1.52	0.79	
Average	1.00	1.02	1.00	1.01	1.02	1.01	1.10	1.02	1.03	1.07	1.03	
Fold change	Mettl3	Mettl14	WTAP	FTO	ALKBH5	YTHDC1	YTHDC2	YTHDF1	YTHDF2	YTHDF3	HNRNPA2B1	
Ang II	1.25	2.08	1.06	1.17	7.46	1.65	1.27	0.58	0.65	0.43	0.86	
	1.50	1.69	1.44	1.32	6.76	1.58	1.09	0.72	0.63	0.49	1.02	
	1.45	1.16	1.17	1.55	5.47	3.86	1.88	0.81	0.71	0.50	1.38	
Average	1.40	1.64	1.22	1.35	6.56	2.36	1.41	0.71	0.66	0.47	1.09	
P value	0.01	0.10	0.15	0.14	0.00	0.15	0.51	0.10	0.11	0.09	0.81	

-Figure 2D: please show all blots used for quantification of this data. The current blots do not appear to have 2x protein shifts.

Response:

Thank you for your kind suggestion. We have cropped the image again to avoid displaying the non-specific band larger than 44 Kd in the Figure 2. As your comment, we showed all un-cut western blots used for quantification in the below Response Figure 3.

Response Figure 3: Representative original images of ALKBH5 expression by western blot in cardiac Td+ cells from *Cx3cr1Cre^{Td}* mice with and without Ang II infusion.

-Figure 2E: Reporting only percentage data is very difficult to interpret. It appears that there is a dramatic overall reduction in tdTomato+ cells. Does *Alkbh5* deletion in macrophages result in changes in cardiac macrophage numbers in the steady state or following challenge? Which subsets of macrophages are most influenced by *Alkbh5*-deletion? What about blood monocytes?

Response:

As shown in the Figure 6A, there were not significant difference of Td+ cells in hearts from C57BL6/J mice transplanted bone marrow from *ALKBH5^{macKO}-Td-BM* and *Cx3cr1Cre-Td-BM* mice. Actually, ALKBH5 deletion did not change Td+ cells in both steady state or following challenge. We apologize for this unrepresentative image in the revised Figure 2G, and provided a more typical image of SMA+Td+ cells in the

hypertensive heart from *ALKBH5^{macKO}-Td* mice. We then quantified the cardiac Td+ cells in the steady state or following challenge. The results showed increased Td+ cells after Ang II infusion, but *ALKBH5^{macKO}* had no effect on cardiac macrophage numbers. In fact, we have detected blood monocytes, and found *ALKBH5^{macKO}* had no effect on blood monocytes. We added this data in the following (Response Figure S4). Ang II-infusion increased blood CD11b+Ly6c^{hi} monocytes, and ALKBH5 deficiency did not change blood monocyte numbers.

Response Figure 4: Specific deletion of ALKBH5 in Cx3cr1 lineage had no effect on circulating blood monocytes. Representative images of blood Ly6C+CD11b+ cells (n =5). Error bars indicate mean ± SEM. n.s. indicates nonsignificant.

-It would be helpful to understand the macrophage response following IL-11-OE. Data for macrophage numbers and phenotype by flow cytometry should be shown for the full experiments in control and challenged mice (WT, IL1OE, KO, KO IL1OE), such as in Figure 6E/F.

Response:

Thank you for your comments. According to the reviewer's suggestion, we added CD11b and SMA FACS data in Td+ cells. FACS analysis indicated that ALKBH5 deficiency or IL11 overexpression did not influence cardiac Td+ cells infiltration. Consistent with the Masson staining and western blot data in Figure 6E/F, the flow cytometry data also showed that IL11 overexpression reversed the decreased percentage of cardiac SMA+CD11b+ in Td+ cells in C57BL6/J mice transplanted with BM from *ALKBH5^{macKO}-Td* mice. The new data were added in the Figure 6A, and the revised text is lines 4-6 at page 12.

-Figure 4: It is surprising that a short-term parabiosis study would incite such dramatic changes in the heart injury model, given the expected minor contribution of chimeric cells to the cardiac myeloid pool. Previously published studies suggested that ~30% of circulating monocytes will come from the parabiont and since the majority of tissue resident macrophages are long-lived/self-maintained, it's surprising that much more than a few percentage of cardiac macrophages would be from the donor. Could the authors discuss how such a minor proportion of cells might inhibit cardiac inflammation? It seems like this experiment would be dramatically more likely to work in a CCR2^{-/-} pair, where all circulating monocytes could come from the donor.

Response:

Thank you for your kind suggestion. As we response to Reviewer 2, we supposed that macrophage-to-myofibroblast transition is working in combination with fibroblast activation in the cardiac fibrosis process. ALKBH5/IL-11 pathway activation is mainly existed in cardiac macrophages. Then the secreted IL-11 induces macrophage-to-myofibroblast transition, as well as promotes cardiac fibroblast activation via directly binding to the receptor IL11RA1 in both macrophages and fibroblasts, since we detected IL11RA1 upregulation in both cells after AngII treatment (Response Figure 2 and Figure 5J).

As your comments, we utilized *CCR2 KO* mice (Figure S8A), which lacked peripheral blood CD1b⁺Ly6c^{hi} monocytes (Figure S8B), to perform parabiosis experiments (Fig. 4A). Ang II-infusion increased chimeric Td⁺ cells in hearts of *CCR2 KO* mice. However, ALKBH5 KO in donor did not affect Td⁺ cell number in the recipient *CCR2 KO* mice (Fig. 4B). Importantly, FACS analysis showed that *CCR2^{KO}* mice conjoined with *Cx3cr1Cre; ALKBH5^{fl/fl}; Rosa26^{Td}* mice displayed decreased percentage of CD11b+SMA⁺ cells in Td⁺ cells compared to *Cx3cr1Cre; Rosa26^{Td}; ALKBH5^{wt/wt}* mice after Ang II-infusion (Fig. 4B), which were further confirmed by immunostaining of SMA in hearts of conjoined *CCR2^{KO}* mice (Fig. 4C). The recipient mice, that received blood from *Cx3cr1Cre; ALKBH5^{fl/fl}; Rosa26^{Td}* mice, showed decreased E/e' ratio, reduction of cardiac fibrosis, decreased SMA, collagen I and III expression Compared with receiving blood from *Cx3cr1Cre; ALKBH5^{wt/wt}; Rosa26^{Td}* mice after Ang II treatment (Fig. 4D-4H). The new data were added in the Figure 4, The revised text is lines 10-42 at page 9.

Figure 4

Fig. 4. ALKBH5 in circulating monocytes-derived macrophage contributes to hypertension-induced cardiac fibrosis and dysfunction. **A**, Diagram of parabiosis between *CCR2*^{KO} and *Cx3cr1Cre; Rosa26*^{Td} or *Cx3cr1Cre; ALKBH5*^{fl/fl}; *Rosa26*^{Td} mice, respectively. **B**, Representative images and quantification of flow cytometry analyses of CD11b+SMA+ cells gated on Td+ cells in hearts from *CCR2*^{KO} mice cojoined with *Cx3cr1Cre; Rosa26*^{Td} or *Cx3cr1Cre; ALKBH5*^{fl/fl}; *Rosa26*^{Td} mice. Error bars indicate mean ± SEM. n = 5. n.s. indicates nonsignificant. **P<0.01. **C**, Representative immunofluorescent images and quantification of SMA+ cells in Td+ cells of cardiac tissues from *CCR2*^{KO} cojoined with *Cx3cr1Cre; Rosa26*^{Td} or *Cx3cr1Cre; ALKBH5*^{fl/fl}; *Rosa26*^{Td} mice (n=5). Error bars indicate mean ± SEM. Scale bar, 100 μm. *P<0.05. **D-E**, Representative echocardiography images of ejection fraction (D) and E/e' (E) of the *CCR2*^{KO} cojoined with *Cx3cr1Cre; Rosa26*^{Td} or *Cx3cr1Cre; ALKBH5*^{fl/fl}; *Rosa26*^{Td} mice after Ang II treatment for 14 days, with indices of cardiac ejection fraction and E/e' at right. Error bars indicate mean ± SEM. n.s. indicates nonsignificant. **P<0.01. **F-G**, Representative images of Masson trichrome staining (F) and quantification (G) of positive fibrotic area (n = 5). Error bars indicate mean ± SEM. **P<0.01. Scale bar, 100 μm. **P<0.01. **H**, Representative images of SMA and ECM genes collagen I and III in cardiac tissues shown by western blot (n=5). **P<0.01.

-Figure 4: necessary control data is missing that would allow for interpretation of this experiment. A) blood chimerism needs to be reported after chimerism, b) cardiac macrophage replacement from donor cells should be reported in untreated and ANG-II treated mice, and c) flow cytometry needs to be performed to assess changes in macrophage phenotype comparing tdTomato+ and tdTomato-neg subsets within the heart. Again, imaging shows reduced tdTomato+ cells, suggesting that intrinsic changes in macrophage numbers may be playing a role in the activation of SM cells.

Response:

Thank you for your suggestive comments. As your comments above, we performed new parabiosis experiment by conjoining the generated *CCR2^{KO}* mice with *Cx3cr1Cre; Rosa26^{Td}; ALKBH5^{wt/wt}* or *Cx3cr1Cre; ALKBH5^{fl/fl}; Rosa26^{Td}* mice, and assessed the chimeric Td+ cells in the blood, and ALKBH5 knockout had no effect on blood Td+ cell chimerism (Figure S8C). We then evaluated cardiac Td+ derived cells by FACS analysis, and observed few CD11b+SMA+Td+ cells in hearts from *CCR2^{KO}* mice under steady state. Ang II significantly increased CD11b+SMA+Td+ cells in hearts from *CCR2^{KO}* mice. *CCR2^{KO}* mice conjoined with *Cx3cr1Cre; ALKBH5^{fl/fl}; Rosa26^{Td}* mice displayed decreased percentage of CD11b+SMA+ cells in Td+ cells compared to *Cx3cr1Cre; Rosa26^{Td}; ALKBH5^{wt/wt}* mice (Figure 4B). As your comments, we further assessed CD11b+SMA+ cells from tdTomato-neg subsets within the hearts from *CCR2^{KO}* mice. The results showed few CD11b+SMA+ cells from Td- subsets after PBS or Ang II administration (Figure S8D). The new data were added in the Figure S8.

-Deletion of ALKBH5 leads to decreased macrophage proliferation and reduced activation of SM and Fibroblasts in co-culture assays. These data clearly show that cardiac macrophages rely on ALKBH5 for normal inflammatory function. Furthermore, it makes analysis of the proposed MMT profile of these cell difficult to assess using this model.

Response:

These are excellent questions. Actually, ALKBH5 deficiency mainly decreased monocyte/macrophage-derived myofibroblast proliferation as shown by the decreased Ki67+Td+ cells in hearts from Ang II treated *Cx3cr1cre; Rosa26^{Td}; ALKBH5^{fl/fl}* mice compared to *Cx3cr1cre; Rosa26^{Td}; ALKBH5^{wt/wt}* mice (Supplemental Figure S5B). Considering your concern about the potential effects of ALKBH5 on macrophage inflammatory function, we further evaluated the inflammatory markers in ALKBH5 deficient macrophages, and found that ALKBH5 knockout had no effect on IL1 β and MCP1 expression under PBS or Ang II treatment. The new data were added in the Response Figure 5.

Response Figure 5: ALKBH5 knockout in macrophages had no effect on IL1 β and MCP1 expression under PBS or Ang II treatment. qPCR analysis of mRNA expression levels of IL1 β and MCP1 in cardiac Td+ cells from *Cx3cr1Cre; Rosa26^{Td}* and *Cx3cr1Cre; ALKBH5^{fl/fl}; Rosa26^{Td}* mice. Error bars indicate mean \pm SEM. n=3. n.s. indicates nonsignificant. **P<0.01.

-Figure 5A: Please report adjusted P-value and FDR for macrophage RIP—seq study. Also, unbiased analysis and full datasets from the gene expression analysis need to be shared. Top 10 pathways enriched would also help to understand the macrophage response to ANG-II.

Response:

Thank you for your suggestions. We provided the adjusted P-value and FDR in the new full datasets of the RIP-sequencing in the Supplementary Dataset (RIP-sequencing Dataset). Our ALKBH5 RNA immunoprecipitation-sequencing (RIP-seq) data revealed that several genes are the direct targets of ALKBH5. We then performed pathway enrichment analysis to show the pathway regulated by ALKBH5 responsive to Ang II. Among the top 10 enriched pathways, we observed that Tgf-beta signaling pathway was associated with ALKBH5, which further indicated that ALKBH5 in macrophages regulated Ang II induced MMT. The new data were added in Figure 5B, and the revised text is lines 7-11 at page 10.

- Figure S7: Please also show control and knockout data for mice that were not treated with ANG-II. Gating approach is disturbing and brings into question the gating used across the manuscript. Are the authors solely generating flow cytometry gates based on tdTomato-negative samples? While this would be potentially acceptable for antibody staining, it is a mistake for reporter mice, particularly when using inflammatory models. This is because cells within tissue often bleb or apoptosis, and small cellular components can be taken up by neighboring cells (this is not restricted to primary phagocyte lineages). Thus, a modest level of tdTomato can often be detected in cells that are indeed tdTomato-negative. There is an obvious cut-off between the 3-4 log of the data where it seems to be much more appropriate to perform this analysis. Did CX3CR1-creERT2 ALKBH5-flox mice show similar tdTomato+ and tdTomato-macrophage numbers compared with controls? It would be expected that a proliferation defect in the ALKBH5-deleted cells may lead to dramatic replacement of these cells during the “tamoxifen-rest” period, which would be evident by a reduced number of Tomato-cells in the ALKBH5-deleted hearts, even in the absence of injury.

Response:

Thank you for your excellent suggestions. We performed new FACS, Masson trichrome staining and echocardiography of *Cx3cr1Cre^{ERT2}; Rosa26^{Td}* and *Cx3cr1Cre^{ERT2}; Rosa26^{Td}; ALKBH5^{flox/flox}* mice with PBS treatment. There were not significant difference of fibrosis and cardiac function between *Cx3cr1Cre^{ERT2}; Rosa26^{Td}* and *Cx3cr1Cre^{ERT2}; Rosa26^{Td}; ALKBH5^{flox/flox}* mice under steady state.

As your kind suggestions, we gated the tdTomato positive and negative clusters at 3-4 log of FACS data. As the previous study, Td+ mainly labeled the cardiac Lyve1+

resident macrophages. However, ALKBH5 deletion had no significant effect on Td+ derived Lyve1+ cell cluster under both steady and hypertensive state. We then gated CD11b+ cells in Td- cells and also observed no change of CD11b+Td+ cells between *Cx3cr1Cre^{ERT2}; Rosa26^{Td}* and *Cx3cr1Cre^{ERT2}; Rosa26^{Td}; ALKBH5^{flx/flx}* mice (Figure S7B). The new data were added in the Figure S7.

Figure S7

Figure S7: ALKBH5 deletion in cardiac resident macrophages has no effects on MMT and cardiac fibrosis and dysfunction. **A**, Diagram of deletion ALKBH5 in cardiac resident macrophages. **B**, Representative flow cytometry analyses of cardiac CD11b+SMA+ cells gated on Td+ cells from *Cx3cr1Cre^{ERT2}; Rosa26^{Td}* and *Cx3cr1Cre^{ERT2}; ALKBH5^{flx/flx}; Rosa26^{Td}* mice, with representative images at left and quantification at right. Error bars indicate mean \pm SEM. n=5. n.s. indicates nonsignificant. **C**, Representative images of Masson trichrome staining in cardiac tissue and quantification of positive fibrotic area. Error bars indicate mean \pm SEM. Scale bar, 100 μ m. n.s. indicates nonsignificant. **D-E**, Representative echocardiography images and quantification of ejection fraction (D) and E/e' (E) of above mice. Error bars indicate mean \pm SEM. n.s. indicates nonsignificant.

-LNP assays should show biodistribution and whether the therapeutic has effects in other tissues and cell types.

Response:

Biodistributions of injected LNPs in mice were measured by fluorescence bioimaging at 24 hours after administration. Fluorescence bioimaging showed that LNPs mainly

accumulated in the heart and liver (Response Figure 6A). Since LNPs had no significant effects on the liver fibrosis (Response Figure 6B), we did not show the distribution of LNPs in the liver. And in order to better show the effects of siRNA-LNP on heart, we focused on the heart for photography.

Response Figure 6

Response Figure 6: **A**, Representative fluorescence imaging combined with microCT after intravenous injection of DiR-labeled C12-200 lipid nanoparticles. **B**, Representative images of positive fibrotic area of Masson trichrome staining in liver of mice with scramble or ALKBH5 siRNA/LNP (n=5). Error bars indicate mean \pm SEM. Scale bar, 100 μ m. **P<0.05.

Minor Comments:

-overall the writing needs to be toned down. The authors often write “confirm, show, validate, etc” language, when it is more appropriate to say “suggests, supports, or infers”. An example is when discussing pseudotime trajectory analysis between clusters (Line 55), the data “suggests” there may be a link between macrophage and a fibroblast subset. It does not show that one exists, and the authors should emphasize this point to justify why it needs to be tested rigorously in multiple experimental models. In addition, many of the data shown have multiple interpretations and should be included in the paper – this is particularly true with MMT data. The majority of MMT data would also support that a subset of SM cells are upregulating macrophage-associated markers.

Response:

Thank you for your kind comments. We toned down throughout writing by changing “confirm, show, validate, etc” into “suggests, supports, or infers”.

Thank you for your suggestions very much. We revised the results of pseudotime trajectory as “we then used scVelo to visualize RNA velocity, and the pseudotime trajectory inferred that there may be a link between circulating monocytes-derived macrophages and myofibroblasts”. The revised text is in lines 31-33 at page 3.

We also excluded the interpretations that SM cells might upregulate macrophage-associated markers in the result section. The revised text is lines 2-15 at page 4.

-Sex as a biological variable is not discussed appropriately.

Response:

We used only male in our study. The revised text is in lines 5-6 at page 17 in Methods.

-Figures need to be reported in order they are presented in the text. An example is that

Fig 2D is discussed before 2A-C.

Response:

Thank you for your kind comments. We ordered the Figure reported in our manuscript.

-Authors claim to have sorted non-cardiomyocytes for scRNA-seq. However, in the methods they simply filtered cells through a 40um strainer. Please describe what was actually performed in the text. “Cells were filtered through a 40um filter to enrich for non-CM cells”. To state the cells were sorted mislead the reader and also confuses us considering that CMs are present in the scRNA-seq analysis (Fig 1A).

Response:

Thank you for your comment. Actually, the non-CM cells were collected through a 40 um filter. We have revised the text at line 5 at page 22 in Methods section.

REVIEWERS' COMMENTS

Reviewer #1 (Remarks to the Author):

the authors have addressed all my comments.

Reviewer #2 (Remarks to the Author):

The authors have provided adequate and specific responses to the initial series of comments. The impact of the paper in current form is improved and the reviewer has no further concerns.

Reviewer #3 (Remarks to the Author):

I greatly appreciate the efforts taken by the authors to address my concerns with the original submission. These additions have improved the rigor of the manuscript and I have no additional comments.